# Satellite retrieval of cloud base height and geometric thickness of low-level cloud based on CALIPSO

Xin Lu[1], Feiyue Mao[1, 2, 3], Daniel Rosenfeld[1, 4], Yannian Zhu[5, 6, *], Zengxin Pan[4], Wei Gong[7]

[1]State Key Laboratory of Information Engineering in Surveying, Mapping, and Remote Sensing, Wuhan University, Wuhan, 430079, China

[2]School of Remote Sensing and Information Engineering, Wuhan University, Wuhan, 430079, China

[3]Collaborative Innovation Center for Geospatial Technology, Wuhan, 430079, China

[4]Institute of Earth Sciences, The Hebrew University of Jerusalem, Jerusalem, 91904, Israel

[5]School of Atmospheric Sciences, Nanjing University, Nanjing, 210023, Chinas

[6]Joint International Research Laboratory of Atmospheric and Earth System Sciences & Institute for Climate and Global Change Research, Nanjing University, Nanjing, 210023, China

[7]Electronic Information School, Wuhan University, Wuhan, 430072, China

*Correspondence to*: Yannian Zhu (yannianzhu@gmail.com)

**Abstract.**

Satellite-based cloud base and top height (CBH and CTH) and cloud geometrical thickness (CGT) are validated against ground-based lidar measurements and provide new scientific insights. The satellite measurements are done by the Cloud Aerosol Lidar and Infrared Pathfinder Satellite Observation (CALIPSO). The retrieval methodology is built on the 333-m resolution low-level water cloud data obtained from the Vertical Feature Mask product of CALIPSO. The methodology is based on the definition that CBH of boundary layer clouds is the lowest cloud base over an area of several tens of km. This allows taking the CBH of the neighbouring penetrable shallower cloud as having CBH representative for the entire cloud field. The methodology over the ocean was validated based on observations from two surface-based ceilometer measurements in the islands of Barbados and the Azores, with an error standard deviation of ±115 m. Validation over land was based on 4 years data of 138 terrestrial ceilometer sites with an error standard deviation of ±220 m. The unprecedented accurate CBH allows us to obtain CGT, which is an essential parameter in the understanding of the aerosol-cloud interaction. Based on this newly developed methodology, we retrieved the annual, seasonal, and diurnal distributions of global CBH, CTH, and CGT for two years, and analysed the variations of CBH and CTH over the ocean and land. Climatology of the annual mean cloud geometrical properties show that: (1) The lowest CBH occurs over the eastern margins of the subtropical oceans and increases westward from 300-400 to 800-900 m. The CGT increases from 300 to 1200 m, respectively. In the western part of the tropical oceans, CBH is 500-600 m and CGT is ~1500 m. (2) A narrow band of lower CBH and CGT occurs over the Equator, especially over the eastern parts of the oceans. (3) CBH and CGT over the tropical rain forests (Amazon and Congo) are 1200 and 1500 m, respectively. CBH over the drier tropical land is 1500-2000 m, with CGT of 800-1000 m. (4) Lowering CBH towards Antarctica in the Southern Oceans, while deepening CGT. (5) Seasonally, the mid-latitude global oceans have the lowest CBH (mostly below 500 m) and CGT in summer seasons, and the highest values in winter. (6) There is just an obvious difference

between day and night for the maximum CTH and CGT over the tropics. Over the ocean, there is no discernible difference in
CBH, but during night CTH is higher by ~300 m.

## 1 Introduction

Satellite retrievals of cloud base height (CBH), cloud top height (CTH), and cloud geometrical thickness (CGT) are essential
for quantifying cloud dynamic and microphysical properties (Rosenfeld et al., 2016;Zhao et al., 2012). Atmospheric aerosols,
which serve as cloud condensation nuclei (CCN), control the size and number concentration of cloud droplets and regulate the
radiation balance of the Earth-atmosphere system (Rosenfeld et al., 2019;Twomey, 1977;Albrecht, 1989;Garrett and Zhao,
2006). Satellite retrieval of CCN depends on the cloud base updraft (Rosenfeld et al., 2016;Efraim et al., 2020;Zheng, 2019),
which is linearly related to CBH (Zheng and Rosenfeld, 2015;Zheng et al., 2020). The cloud base droplet concentration ($N_d$)
is determined by the cloud base updraft, CCN, supersaturation, wind shear, and so on, which in turn determines the cloud's
albedo for a given liquid water path (Twomey, 1974;Sato and Suzuki, 2019). However, the current satellite-retrieved $N_d$
requires to assume an adiabatic fraction ($f_{ad}$) of the cloud water, which is usually taken as $f_{ad} =1$ (Merk et al., 2016;Grosvenor
and Wood, 2014). In reality, $f_{ad}$ is often much smaller than 1, which leads to a serious underestimation relative to the in situ
measured $N_d$ (Efraim et al., 2020). Accurate information on cloud base and cloud thickness is a necessary condition for retrieval
of adiabatic fraction. Therefore, accurate CBH and CGT are extremely important to reduce the uncertainty of aerosol-cloud
interaction.
CBH has practical significance for the aviation community (Noh et al., 2017). Recent studies have shown that CGT can isolate
the aerosol-cloud interaction from the influence of meteorology (Rosenfeld et al., 2019;Sato and Suzuki, 2019). CBH and CTH
are fundamental cloud properties that are required to be parameterized correctly for improving model simulations of climate
and climate change (Grosvenor et al., 2017;Zhao and Suzuki, 2019;Lenaerts et al., 2020;Ma et al., 2018). Therefore, it is
necessary to obtain the accurate CBH and CTH and further retrieve CGT. All these properties are important to understand the
complex cloud microphysical processes and aerosol-cloud interaction (Stephens and Webster, 2010;Dupont et al., 2011;Kyle
et al., 2016). Low-level clouds reflect most of the incident solar radiation received by the Earth back to space, and they are of
great interest for various applications (such as retrieval of cloud microphysical properties, weather prediction, and so on).
Therefore, high-precision CBH and CGT data of low-level clouds are the foundation of the follow-up aerosol-cloud interaction
research.
Satellites provide a wide range of cloud observations from space (Stephens et al., 2019). It is feasible to retrieve CTH based
on satellite data because satellites can observe the cloud top directly (Weisz et al., 2007). Although there is often a large
uncertainty in the cloud top heights obtained from passive satellite observations, it is relatively simple to retrieve. In contrast,
the retrieval of CBH is much more challenging but necessary for retrieving CGT. There are already many different methods
to retrieve the CBH based on different satellite observations. The Suomi National Polar-orbiting Partnership (Suomi NPP)
Visible Infrared Imaging Radiometer Suite (VIIRS) retrieves CBH based on the CTH and CGT. However, VIIRS does not

directly observe CGT, which is calculated by assuming $f_{ad} = 1$ (Baker, 2011). To investigate the accuracy of VIIRS CBH retrieval algorithm, Seaman et al. (2017) compared the CBH from the VIIRS with those from the CloudSat cloud profile radar. They showed that because the VIIRS official retrieval algorithm is insensitive to upper clouds, the CBH error for all clouds in global is 3.7 km, and even for clouds with accurate CTH, the root-mean-square error (RMSE) of CBH reaches 2.3 km. Böhm et al. (2019) retrieved global CBH data based on multi-angle satellite data, and the validation results based on ground-based observations showed that the RMSE of CBH obtained by this method was ~400 m. Li et al. (2013) conducted the retrieval of global marine boundary layer CBH based on boundary layer lapse rate observation from the A-train satellite constellation. By comparing their retrieval to CloudSat CBH retrieval, a standard deviation of 540 m was found. Zhu et al. (2014) used the imager of the Suomi NPP VIIRS and retrieved the cloud base of convective clouds at an accuracy of 200 m, but this retrieval relied on the strong contrast between the cloud and underlying surface brightness, and could not work at night. CloudSat is an essential active cloud radar observation satellite. However, CloudSat has difficulties retrieving the CBH of low-level clouds for the following reasons: a) The ground clutter prevents detection of a very low base. b) Rain from precipitating clouds produces radar returns below cloud base. c) Due to the dependence of radar reflectivity on the 6$^{th}$ power of cloud droplet diameter, the reflectivity of clouds with small droplets can be below the CloudSat minimum detectable signal, especially near cloud base where cloud droplets are smallest. It can be seen that these CBH retrieval methods either have low accuracy or do not provide all-day CBH data. Therefore, there is a yet unfulfilled scientific need to obtain high-precision all-day CGT/CBH based on active satellite observations.

Satellite lidars, such as the Cloud Aerosol Lidar and Infrared Pathfinder Satellite Observation (CALIPSO), have the potential for accurate retrieval of CBH (Winker et al., 2009). However, CALIPSO typically provides only CBHs for thin clouds, because it can penetrate only clouds with an optical thickness of less than 5 (Mace and Zhang, 2014). When the thickness of the cloud is sufficient to fully attenuate the CALIPSO lidar signal, CALIPSO cannot provide information about the base of these clouds. Mülmenstädt et al. (2018) retrieved the global CBH using CALIPSO Vertical Feature Mask (VFM) data and evaluated the retrieval algorithm based on ground-based ceilometer observation from about 1500 stations across the continental USA. They extrapolated CBH information from a surrounding field onto profiles for which the lidar signal was attenuated using CALIPSO's VFM, and took the mean of all considered VFM CBH retrievals within a distance of 100 km weighted by estimated uncertainties to determine the CBH at a given point of interest, but their overall RMSE of CBH exceeded 500 m. This provided the basic idea and motivation to retrieve the CBH at a higher precision in this study. This basic idea is that the CBH of the optically thin clouds can be used as the CBH for the whole scene at a given range (~100 km). There are many other challenges. For example, strong surface echoes can affect the identification of cloud bases of CALIPSO observations (Burton et al., 2013). In addition, in aerosol-prone regions, such as East Asia, South Asia and desert regions, due to the influence of aerosols in the boundary layer, the low-level cloud may be masked by dense aerosol layers, thereby affecting the determination of the cloud layer (Vaughan et al., 2005). Further, large areas of elevated cloud layers can also interfere with the CBH retrieved by active CALIPSO observations. These factors are expected to result in a large uncertainty in the typical CBH obtained directly based on CALIPSO observations.

To solve the above problems, we derived a new methodology by using the highest resolution of CALIPSO measurements to retrieve the global distribution of CBH, CTH, and CGT of low-level clouds and validate against in situ ceilometer measurement. The data used in this study are presented in Section 2 and the retrieval method is given in Section 3. The CALIPSO-retrieved CBHs are evaluated and validated against in situ ceilometer measurements (Section 4). Based on the validated CBH, we retrieved CTH and CGT globally and produced global annual, seasonal and diurnal distribution maps of CBH, CTH, and CGT

(Section 5). Specific spatial patterns are further discussed in Section 6. Conclusions are provided in Section 7.

## 2 Data

### 2.1 CALIPSO VFM data

The satellite data analysed in this study are from the Cloud Aerosol Lidar with Orthogonal Polarization (CALIOP) lidar on CALIPSO satellite, which can provide two-dimensional (vertical and horizontal along the satellite track) information of clouds

with global coverage (Winker et al., 2007). CALIPSO, jointly developed by NASA and CNES, is a sun-synchronous orbiting satellite with an orbital inclination of 98.2°, an orbital altitude of 705 km, a revisit period of 16 days, and an equatorial crossing time of approximately 13:30 local time. The cloud top and base can be obtained from the CALIOP VFM product (Winker et al., 2009). For each CALIOP attenuated backscattering profile, the VFM product identifies features classified as clouds, aerosols, stratospheric features, and surfaces; this is known as feature type. The VFM also provides the thermodynamic phase

of cloud layers (water cloud, ice cloud) and horizontal resolution (333 m, 1 km, 5 km, 20 km, 80 km) that the retrieval was based on. The CALIOP retrieval algorithm must average over a horizontal distance to collect sufficient signal that allows the identification of features on the background noise of atmospheric molecules scattering. The official CALIPSO classification algorithm suffers from the misclassification of clouds and aerosols at low resolution (Mace and Zhang, 2014;Vaughan et al., 2005). In this study, we use VFM version 4.10 data for the full years of 2014 and 2017. The VFM files are available from

ASDC (https://eosweb.larc.nasa.gov/). To ensure that high-quality CALIPSO VFM data is used, we limit the VFM quality assurance flag as "high" (Mülmenstädt et al., 2018).

### 2.2 Ground ceilometer data

The retrieval algorithm is developed and validated using ground-based ceilometer observations. To represent the different types of low-level clouds around the world, we used ceilometer sites located at different latitudes over the ocean and land (two

marine sites and 138 continental sites) respectively to validate the CALIPSO-retrieved CBH. One marine site is at the low latitude and one at mid-latitude. The low latitude marine site is Barbados (13.2° N, 59.4° W) (https://barbados.mpimet.mpg.de/). The temporal resolution of the ceilometer at Barbados site is 10 seconds, and the vertical resolution is 15 m. The mid-latitude marine site is the Eastern North Atlantic (ENA) site located at the Azores (39.1° N, 28.0° W), operated by the Atmospheric Radiation Measurement (ARM), (https://www.arm.gov/data). At the ENA site, the ceilometer has a temporal resolution of 16

seconds, a vertical resolution of 30 m, and a maximum detection range of 7700 m. The data period used in this study of these

two marine sites is from January 2017 to December 2017, and the cloud types are mostly marine stratocumulus and trade wind cumulus.

Over land, 138 ceilometer sites are located at the southern Great Plains of the USA. These ceilometer data are derived from Automated Surface Observation System (ASOS). The data source is https://mesonet.agron.iastate.edu/request/download.phtml, which is maintained by Iowa Environmental Mesonet of Iowa State University. The data period used in the validation experiment is from January 2017 to December 2020. The ASOS uses a laser beam ceilometer with a time interval of 5-30 minutes, with a vertical resolution of ~30 m, and a vertical detection range of ~3700 m. To ensure the data quality of the ASOS ceilometer observations, we only use ceilometer data with cloud base heights less than 3000 m. In our study, ceilometer observations from marine sites were used in the development of the CBH retrieval algorithm and data from terrestrial sites were used as independent datasets for the validation and further refinement of the algorithm.

### 2.3 Data matching

Figure 1 illustrates the data matching between CALIPSO and the ceilometer site. To obtain the CALIPSO-retrieved CBH that matches the ceilometer-measured CBH, a scene of 1° along the CALIPSO track is selected (the gray shaded area in Figure 1), centered in time on the overpass time and extending 0.5° to the backward and forward along the CALIPSO track. The scene is selected if the distance from the CALIPSO center point to the ceilometer site is less than 150 km, and is used for matching the CALIPSO with the ceilometer data. Then we obtained the distribution of the base height of cloud features observed by ceilometers within 30 minutes before and after the CALIPSO overpass time ($CB_{ceilo}$). To avoid the underestimation of low CBHs and overestimation of high CBHs by ceilometer due to the influence of developing higher-level clouds and ceilometer measurement noise, the lowest 10 % quantile of the $CB_{ceilo}$ is determined as true CBH (Wang et al., 2018).

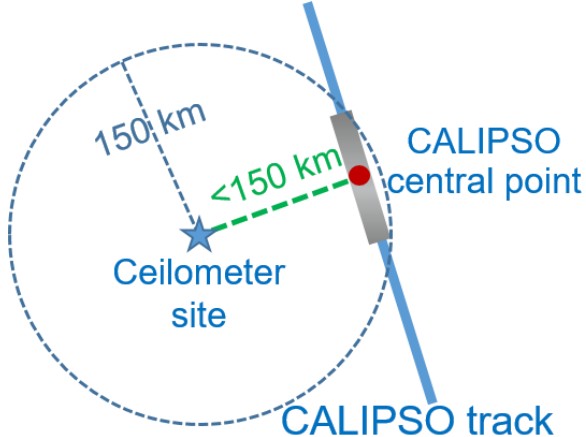

**Figure 1: Schematic of matching between the CALIPSO and a marine ceilometer observation site.** The blue pentagram indicates the location of the ceilometer site, the blue solid line is the CALIPSO track, the red solid circle represents the CALIPSO central point, the dashed

green line represents the shortest distance from the ceilometer site to the CALIPSO central point, and the grey shaded area is the 1° CALIPSO scene. The blue circle is centered on the ceilometer site and has a radius of 150 km.

The retrieval methodology in this study relies on the assumption of the undisturbed boundary layer with a similar cloud base height within the scene. The topography of the islands at the Azores has volcanic peaks with heights up to 500 m, and may violate the homogeneity assumption. To avoid the anomalous uplift of clouds by the topography of these adjacent islands, we restricted the CALIPSO data according to the terrain and wind directions as showed in Figure A1. If the elevation corresponding to the CALIPSO data matching the ENA site ceilometer data is higher than 30 m, the CALIPSO data are rejected. Similarly, to avoid the situation that the clouds traveled above the adjacent islands were perturbed, we rejected the CALIPSO data with the wind direction of southwest/southeast if the CALIPSO track is located at the east/west ocean of ENA site (Figures A1a and A1b). The wind direction is from the meteorological observation data of ENA site, and the elevation data are from the CALIPSO VFM product.

## 3 CALIPSO CGT retrieval algorithm

The objective of the CBH retrieval is to retrieve the forming level of clouds. In the case of a well-mixed boundary layer, it is the lifting condensation level. When clouds are decoupled, that level is usually higher. That formative cloud base level is similar in areas with similar thermodynamic structure, which is conducive to a nearly constant cloud base height. Thus, the CBH of optically thicker clouds that cannot be penetrated by CALIOP can be expressed by the CBH of the surrounding penetrable thin clouds. The retrieval algorithm relies on this assumption, by adopting the lowest reliably detected cloud height along a CALIPSO track of approximately 100 km (1° along the track) as the cloud base height.

### 3.1 Extraction of 333-m horizontal resolution low-level cloud feature

In this study, we retrieve CBH and CGT for CALIPSO VFM scenes which are identified as low-level water clouds. Low clouds are defined following the International Satellite Cloud Climatology Project as clouds distributed below 680 hPa (Hahn et al., 2001), which also complies with the detection range of the ceilometer. Figure 2 displays an example of CALIPSO low-level cloud feature determination. For each 1° scene along the CALIPSO track, the distribution of the low-level water clouds which had sufficient signal to be detected with a horizontal resolution of 333-m (light blue areas in Figure 2c) was obtained. These cloud features were identified based on the CALIPSO feature type data (Figure 2a) and resolution data (Figure 2b). Then, based on the low-level cloud information in Figure 2c, we could retrieve the CBH, CTH, and CGT of this scene.

Using 333-m resolution cloud feature information allows a better separation between the clouds and boundary layer aerosols, because the aerosol identification is based mostly on 1-km or lower resolution data, as evident in Figure 2b (Vaughan et al., 2005). Moreover, CBH obtained from higher resolution VFM data (such as 333-m resolution) was closer to the lifting condensation level (Seung-Hee et al., 2017). Also, low horizontal resolution is most likely to lead to false detection of clouds (Mace and Zhang, 2014). Therefore, it is more reliable to use the water cloud information with the highest resolution of 333

185    m to retrieve the CBH of low-level clouds compared to previous studies based on coarse spatial resolution CALIPSO satellite data. Thus, we chose to use 333-m resolution instead of other resolutions for CBH and CGT retrieval. This is one of the main differences between our algorithm and other current algorithms, such as Mülmenstädt et al. (2018). This largely reduces the impact on CBH retrieval due to official algorithmic misclassification of aerosols and clouds in low-resolution CALIPSO VFM data.

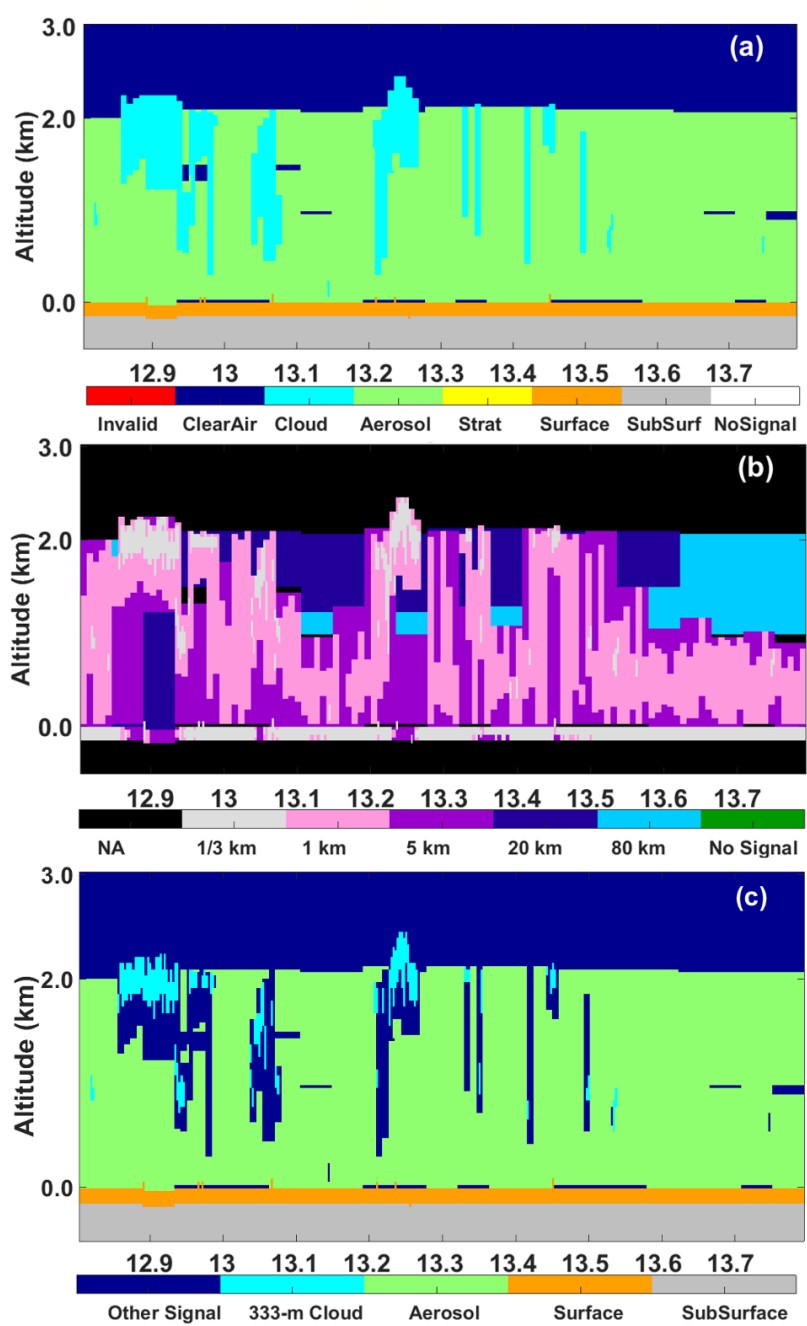

190

**Figure 2: Schematic of CALIPSO low-level cloud feature determination (CALIPSO VFM data at (UTC) 05:51:17 on 3 January 2017).** (a) Latitude-altitude distribution of feature type based on CALIPSO VFM feature type parameter, the light blue areas represent the water cloud features. "Strat" is the stratospheric feature, "SubSurf" is the Sub Surface. (b) Resolution information distribution based on CALIPSO VFM resolution parameter, the gray areas represent the 333-m resolution features. (c) 333-m horizontal resolution water cloud distribution combined by (a) and (b); the light blue areas indicate the 333-m horizontal resolution water clouds, the green areas refer to aerosols at any resolution, the orange area is the surface.

## 3.2 CALIPSO initial CBH in 1° scene

For each 1° scene along the CALIPSO track, based on 333-m horizontal resolution low-level cloud information as described in Section 3.1, we obtained the height distribution of the lowest cloud feature ($H_{min}$) of each water cloud profile under which the surface is detectable as shown in Figure 3a. Then, based on this $H_{min}$ data, we obtained the 10$^{th}$ percentile of the $H_{min}$ distribution for each 1° scene as shown in Figure 3b, which will be used to reduce the interference of strong surface signals and thick aerosols.

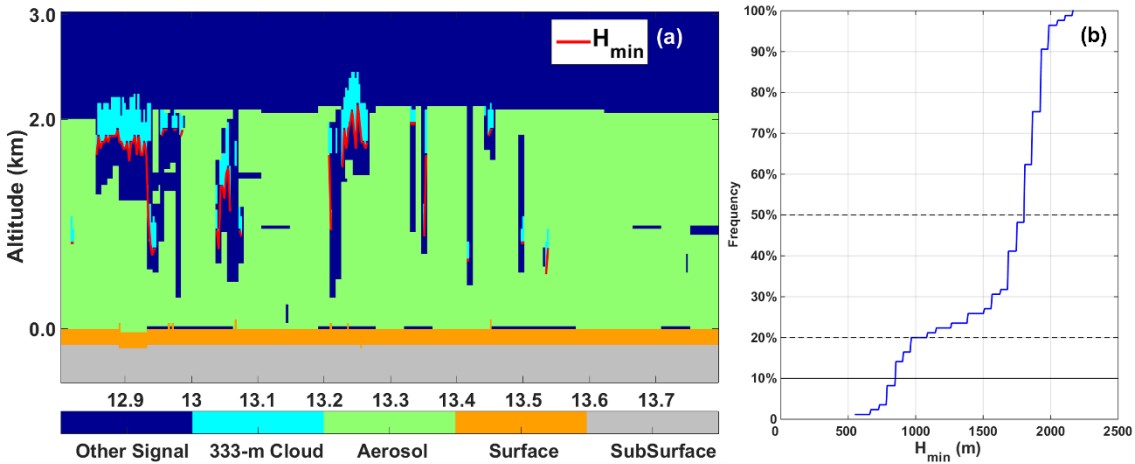

**Figure 3:** (a) The same as Figure 2c but with the distribution of $H_{min}$ (the red line). (b) Cumulative distribution of the $H_{min}$ at any quantile for the scene in Figure 3a (The black dotted lines represent the $H_{min}$ at the 10 %, 20 %, and 50 % quantile, respectively.).

Extremely low $H_{min}$ are more prone to misclassification because the mixture of surface signals and cloud features or VFM misclassified near-surface thick aerosols as clouds (Burton et al., 2013;Vaughan et al., 2005), the lowest height of $H_{min}$ cannot be used as the initial CBH of this 1° scene. To obtain the optimal quantile of the initial CBH, we carried out a sensitivity test based on the CALIPSO-retrieved CBH and the ceilometer-measured CBH from two ground marine observation stations (Barbados site and ENA site) in 2017 as showed in Figure 4. The sensitivity test shows that the application of CALIPSO $H_{min}$ at 10 % quantile as the initial CBH of this 1° scene greatly reduces the problem that the lowest $H_{min}$ retrieved from CALIPSO is much smaller than the true CBH (RMSE reduced by 88 m). Using $H_{min}$ at 10 % quantile as the initial CBH, rather than simply calculating the average CBH of $H_{min}$ as the initial value, goes some way to reducing the effect of high-altitude spreading of water clouds due to convective activity on the retrieval of CBHs.

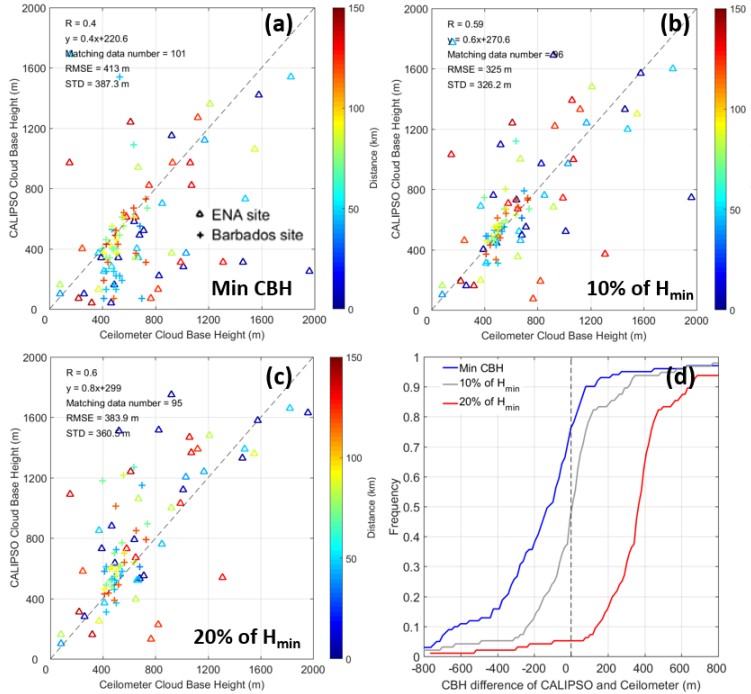

**Figure 4: (a) Scatter plot of CALIPSO CBH (the minimum CBH for each 1° scene) and ceilometer CBH at two marine sites in 2017.** The triangle represents the data for the ENA site, and the crosses represent the data for the Barbados site. The color represents the shortest distance from the CALIPSO ground track to the ceilometer site. R is the Pearson correlation coefficient, y indicates the linear fitting relationship between ceilometer CBH and CALIPSO CBH, matching data number is the data amount of the scatter plot, RMSE is the root-

220 mean-square error and STD is the standard deviation. (b) and (c) are the same as (a), but for CALIPSO CBH at 10 % quantile of $H_{min}$ and 20 % quantile of $H_{min}$, respectively. (d) Cumulative distribution of the difference between CALIPSO CBH and Ceilometer CBH at two sites in 2017.

## 3.3 Determination of CALIPSO CBH, CTH, and CGT of 1° scenes

After getting the initial CBH ($H_{min}$ at 10 % quantile), there are still many factors affecting the determination of the final CBH

in the cloud scenes that contribute to the uncertainty or add limitations to the CBH retrieval. There are many confounding factors, including multilayer cloud fraction ($F_{multi}$), and detection efficiency of CALIOP lidar ($E_{lidar}$ and $E_{lidar\_full}$). These difficulties are overcome by the added selection criteria, which are tested against in situ ceilometer measurements as presented in the following sub-sections.

### 3.3.1 Multilayer clouds

Multilayer status of features can be detected by CALIPSO vertical profile measurements. In VFM data, for each profile (from the surface to the altitude of 20 km), if cloud features are continuous, then it is a profile with a single-layer cloud; if there is more than one cloud segment in that vertical profile, then this profile contains multilayer clouds. In this study, for each 1° CALIPSO scene, only profiles containing continuous single-layer clouds are used to retrieve CBH, CTH, and their thickness, that is, any profile with multilayer clouds is excluded. This is because the aerosol-cloud interaction studies mainly focus on

single-layer clouds. Therefore, when there are too many multilayer clouds in a 1° scene, in order to guarantee the effectiveness of CBH retrieval, this scene will be rejected. We use the $F_{multi}$ to represent the cloud cover of the multilayer clouds in the scene. For a given 1° CALIPSO scene, $F_{multi}$ is calculated using Eq. (1):

$$F_{multi} = N_{multi}/N_{total,} \tag{1}$$

$N_{multi}$ is the number of the CALIPSO lidar profiles that contain multilayer clouds, and $N_{total}$ is the total number of the CALIPSO

lidar profiles collected in a given 1° CALIPSO scene. In order to obtain the optimal threshold of $F_{multi}$, we carried out the sensitivity test as displayed in Figure 5. It can be seen from Figure 5 that when the multilayer cloud fraction limitation is used, the RMSE decreases from 325 m in Figure 4b to ~225 m. Given there are a considerable amount of multilayer clouds on the globe, we chose a moderate multilayer cloud fraction threshold of 40 %. Therefore, when there are too many multilayer clouds in a scene ($F_{multi} > 40$ %), the scene is rejected.

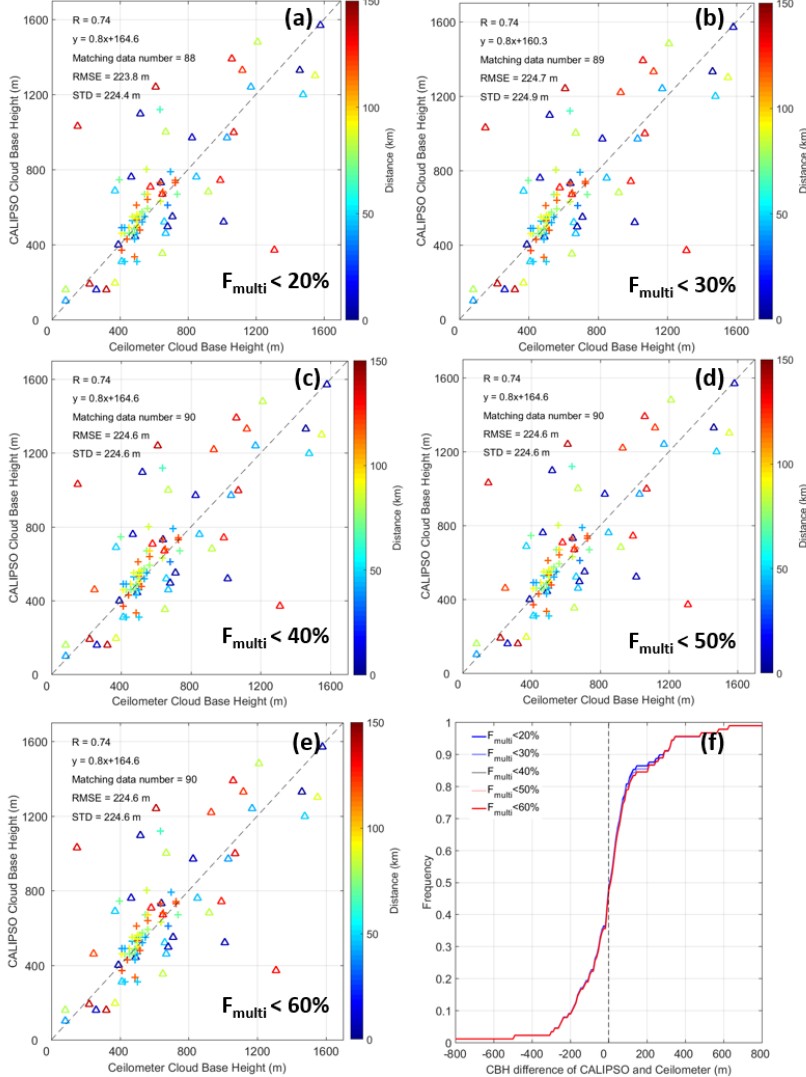

**Figure 5: (a) Scatter plot of CALIPSO CBH and ceilometer-measured CBH at two marine sites in 2017, when selecting scenes with multilayer cloud fraction ($F_{multi}$) < 20 %.** The triangle represents the data for the ENA site, and the crosses represent the data for the Barbados site. The color represents the shortest distance from the CALIPSO ground track to the ceilometer site. (b), (c), (d) and (e) are the same as (a), but selection criteria of $F_{multi}$ in (b) is less than 30 %, (c) is less than 40 %, (d) is less than 50 %, (e) is less than 60 %, respectively. (f) Cumulative distribution of the difference between CALIPSO CBH and Ceilometer CBH at two sites in 2017. Different colored lines represent the cumulative distribution at different selection criteria of $F_{multi}$.

### 3.3.2 Penetration efficiency of CALIOP

When the clouds are sufficiently thick, CALIOP lidar beam cannot penetrate them and reach the surface. Although we can use the cloud base information of thin clouds as a proxy for the cloud base of optically thicker clouds within the field, we still need to consider the penetration efficiency of CALIPSO to thick clouds. The fraction of cloudy pixels in which the lidar penetrates the clouds to their base is defined as lidar penetration efficiency ($E_{lidar}$ and $E_{lidar\_full}$). $E_{lidar}$ is used to determine the lowest penetration efficiency that can still provide valid cloud base information of 333-m resolution cloud in this study, which is calculated using Eq. (2):

$$E_{lidar} = N_{surface\_333}/N_{total\_333,} \tag{2}$$

$N_{surface\_333}$ refers to the number of CALIPSO lidar profiles that have both, 333-m horizontal resolution clouds and a detectable surface, $N_{total\_333}$ is the total number of CALIPSO lidar profiles that detected the 333-m horizontal resolution clouds. The sensitivity test result of $E_{lidar}$ is displayed in Figure 6. We can see that the higher $E_{lidar}$ the better cloud base height retrieval we can get, but when $E_{lidar}$ approaches 1 we lose the ability of detecting CBH of optically thick clouds. Therefore, an optimal $E_{lidar}$ of 50 % was chosen. That is, when the $E_{lidar}$ < 50 % in a scene, in order to guarantee the effectiveness of CBH retrieval, the scene is rejected.

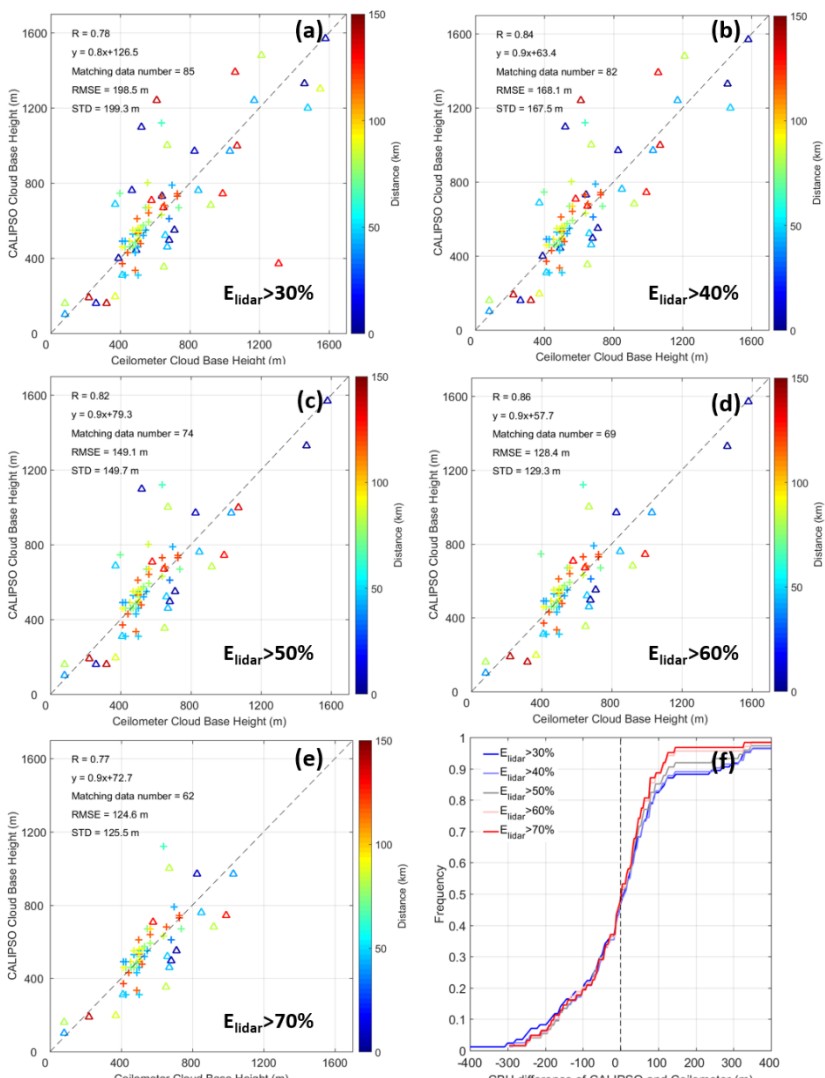

**Figure 6: (a) Same as Fig. 5, but for selecting scenes with penetration efficiency of 333-m horizontal resolution cloud features ($E_{lidar}$) >**
**30 %.** The selection criteria of multilayer cloud fraction is the same as Figure 5c. The triangle represents the data for the ENA site, and the
crosses represent the data for the Barbados site. The color represents the shortest distance from the CALIPSO ground track to the ceilometer
site. (b), (c), (d) and (e) are the same as (a), but selection criteria of $E_{lidar}$ in (b) is larger than 40 %, (c) is larger than 50 %, (d) is larger than
60 %, (e) is larger than 70 %, respectively. (f) Cumulative distribution of the difference between CALIPSO CBH and Ceilometer CBH at
two sites in 2017. Different colored lines represent the cumulative distribution at different selection criteria of $E_{lidar}$.

In addition, the penetration efficiency of full-resolution clouds is also taken into consideration, which is calculated using Eq.
(3):

$$E_{lidar\_full} = N_{surface\_fullCloud}/N_{total\_fullCloud,} \tag{3}$$

$N_{surface\_fullCloud}$ refers to the number of CALIPSO lidar profiles that have both, full-resolution clouds and a detectable surface;

$N_{total\_fullCloud}$ is the total number of CALIPSO lidar profiles that detected the full-resolution clouds. The sensitivity test result of

$E_{lidar\_full}$ is displayed in Figure 7. We can see that the higher $E_{lidar\_full}$ the better cloud base height retrieval we can get, but when
$E_{lidar\_full}$ is greater than 50 %, the amount of matched data is significantly reduced and the ability to retrieval high CBH is lost.
Therefore, an optimal $E_{lidar\_full}$ of 50 % was chosen.

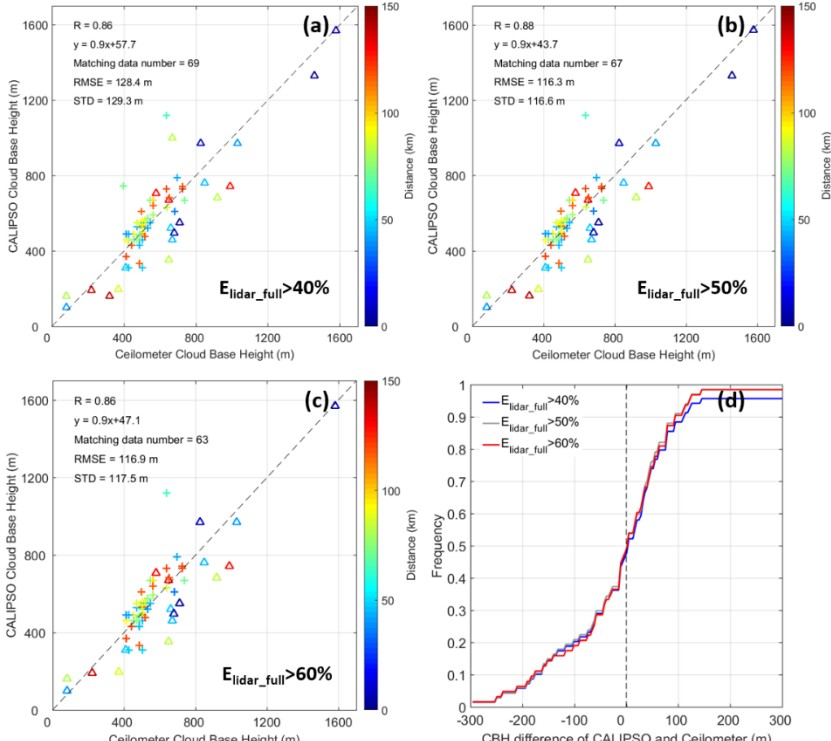

**Figure 7: Same as Fig. 5, but:** (a) when penetration efficiency of full-resolution cloud features ($E_{lidar\_full}$) of one scene is larger than 40 %;
selection criteria of multilayer cloud fraction, and penetration efficiency of 333-m resolution cloud is the same as Figure 6c at two marine
sites in 2017. The triangle represents the data for the ENA site, and the crosses represent the data for the Barbados site. The color represents
the shortest distance from the CALIPSO ground track to the ceilometer site. (b) and (c) are the same as (a), but selection criteria of $E_{lidar\_full}$
in (b) is larger than 50 %, (c) is larger than 60 %, respectively. (d) Cumulative distribution of the difference between CALIPSO CBH and
Ceilometer CBH at two sites in 2017. Different colored lines represent the cumulative distribution at different selection criteria of $E_{lidar\_full}$.

Therefore, after we obtained the initial CBH ($H_{min}$ at 10 % quantile), we reject a 1° CALIPSO scene when:

(a)     The multilayer cloud fraction of this scene is greater than 40 % ($F_{multi} > 40$ %);

(b)     Penetration efficiency of CALIOP lidar of 333-m horizontal resolution cloud features is less than 50 % ($E_{lidar} < 50$ %).

(c)     Penetration efficiency of CALIOP lidar of full-resolution cloud features is less than 50 % ($E_{lidar\_full} < 50$ %).

After the above processing, we obtained the final CBH of that CALIPSO 1° scenes. This cloud base height information is
mainly extracted from broken or thin boundary layer clouds. Then 333-m resolution cloud information as described in Section
3.1 was applied to retrieve the CTH of these scenes. For all water cloud layers with a resolution of 333-m, when the CBH of
low-level clouds is retrieved, we obtain the cloud top height of all 333-m cloud profiles ($H_{max}$) in this scene, and take the mean

height of the highest 10 percentile of $H_{max}$ as the CTH of that scene according to the definition in Zhu et al. (2018). Finally, the CGT of this 1° scene is the difference between CTH and CBH (Scheirer and Macke, 2003).

## 4 Evaluation of CALIPSO-retrieved CBH

### 4.1 Over the ocean

In this study, we used ceilometer-measured CBH data from two ground observation stations at oceanic sites in low and mid latitude (Barbados site and ENA site) in 2017 to obtain sufficient data for the development of CALIPSO CBH retrieval. Finally, 67 sets of matching cases were obtained in 2017, including 20 matching cases for ENA site and 47 for Barbados site (Figure 7b). The statistical analysis of these matching cases shows that the CALIPSO-retrieved CBH has a good consistency with the CBH observed by the ceilometers at these two observation stations. The R is 0.88, the RMSE is only 116.3 m and the standard deviation (STD) is 116.6 m. Due to the restrictions of terrain and wind directions for CALIPSO scenes in ENA site as described in Section 2.2, the ENA site matches fewer cases than the Barbados site in 2017. The cumulative distribution of the CBH difference between CALIPSO and ceilometer in Figure 7d (the gray line) indicates that ~70 % of the matching cases have a deviation of less than 100 m. It can also be seen from Figure 7b (the color represents the distance from the CALIPSO ground track to the ceilometer site) that over the ocean, when the distance is less than 150 km, the deviation between CALIPSO-retrieved CBH and ceilometer CBH has little to do with the distance.

### 4.2 Over land

To validate the applicability over land of the CBH retrieval algorithm, we conducted additional validation experiments using 4 years of continental ceilometer data from 138 sites in the southern Great Plains of the USA (as showed in Figure A2). The data period is taken from 2017 to 2020 because during this period, the ceilometers provide better time resolution of cloud base measurements. Since the cloud base is not as homogeneous over land as over the ocean, we consider using the cloud information below the first peak nearest to the surface in the cloud fraction profile of 1° scenes as a proxy for all cloud base information in this scene (which we defined as the first local peak above the surface). In this way, we avoid missing the newly developed clouds with small sizes. Therefore instead of using $H_{min}$ at 10 % quantile of all clouds as the initial CBH over the ocean, we tested the CBH at different quartiles of the first local peak as the initial CBH for the scene (detailed information is provided in Table B1 in the Appendix). The results show that there is a shallow minimum RMSE when the initial CALIPSO CBH is at the 40 % quantile of the first local peak which closest to surface. Thus on land, the CALIPSO-retrieved initial CBH is 40 % quantile of the first local peak, while over the ocean it is $H_{min}$ at 10 % quantile. Then, based on the CBH data obtained from the above processing, we further tested the effects of $F_{multi}$, $E_{lidar}$ and $F_{lidar\_full}$ over land following the same process as over the ocean, as showed in Figure A3 in the Appendix. From the results, it can be seen that the optimal thresholds for these parameters ($F_{multi}$<40 %, $E_{lidar}$>50 %, and $F_{lidar\_full}$>50 %) on land are consistent with those over the ocean, which also shows

that the CBH retrieval algorithm we developed based on cloud observations from the ocean is applicable on land. These final criteria for CALIPSO CBH retrieval used over the ocean and land are also summarized in Table B2.

As mentioned before due to the complexity of topography and land surface situation, the cloud base height varies at larger
spatial scales. The 150 km distance between the shortest distance from the CALIPSO ground track to the ceilometer site cannot be used for over land validation. We have to shrink the distance to minimize the spatial variability due to the changes over land. We tested the effect of distance (that is the shortest distance from the CALIPSO ground track to the ceilometer site) and observation time on the retrieval results (Figure A4 in the Appendix). The results (Figure A4b) show that the absolute error between the CALIPSO CBH and the ceilometer CBH becomes smaller as the distance decreases and stabilizes at distances
less than 50 km. It is therefore preferable to limit the distance to 50 km for studies on land to better meet the assumptions of a homogeneous CBH within the scene. It can also be seen that the cloud base heights are more evenly distributed during the day-time (Figure A4a, 300-1800 m) than at night, while at night CBHs are mainly concentrated below about 700 m. In addition to the distance limitation, the cloud base homogeneity is further constrained by comparing the lifted condensation level ($H_{LCL}$) to the ceilometer cloud base height. To satisfy the cloud base homogeneity assumption (Efraim et al., 2020), cases are selected
when the absolute difference between the $H_{LCL}$ (calculated from ASOS-observed air temperature and dew point temperature) and ceilometer CBH is less than 200. In summary, the ceilometer measurements over land need to satisfy the following conditions for validating CALIPSO CBH retrieval: 1) The ceilometer is within 50 km radius to the centre of CALIPSO ground track; 2) the ceilometer-measured CBH should have an absolute difference less than 200 m against $H_{LCL}$ as calculated from the surface measured air temperature and dew point.

Ceilometer data that passed these conditions were used for validating the CALIPSO retrieved cloud base height. Figure 8 shows the final verification results over land. Based on 4 years of observations from 138 ceilometer sites in the southern Great Plains, 733 sets of matching cases were obtained in 2017-2020 (day-time: 469 sets; night-time: 264 sets). The statistical analysis of these matching cases (Figure 8a) shows that the CALIPSO-retrieved CBH has a good consistency with the CBH observed by the ceilometers at these continental sites. The R is 0.92, the RMSE is 217.2 m and the standard deviation is 217.1
m. The cumulative distribution of the CBH difference between CALIPSO and ceilometer in Figure 8b indicates that ~70 % of the matching cases have a deviation of less than 200 m. In addition, the day-time results (R=0.92, RMSE=178.0 m) are better than the night-time results (R=0.27, RMSE=273.3 m). From Figure 8e it can be observed that the CBH at night is mainly concentrated below 800 m. This might be due to the effect of low-level clouds and fog patches, which possibly contaminate the ceilometer data. Therefore, it is unreasonable to validate the CALIPSO retrieval against the night-time ceilometer
measurements and day-time data are more suitable for validation.

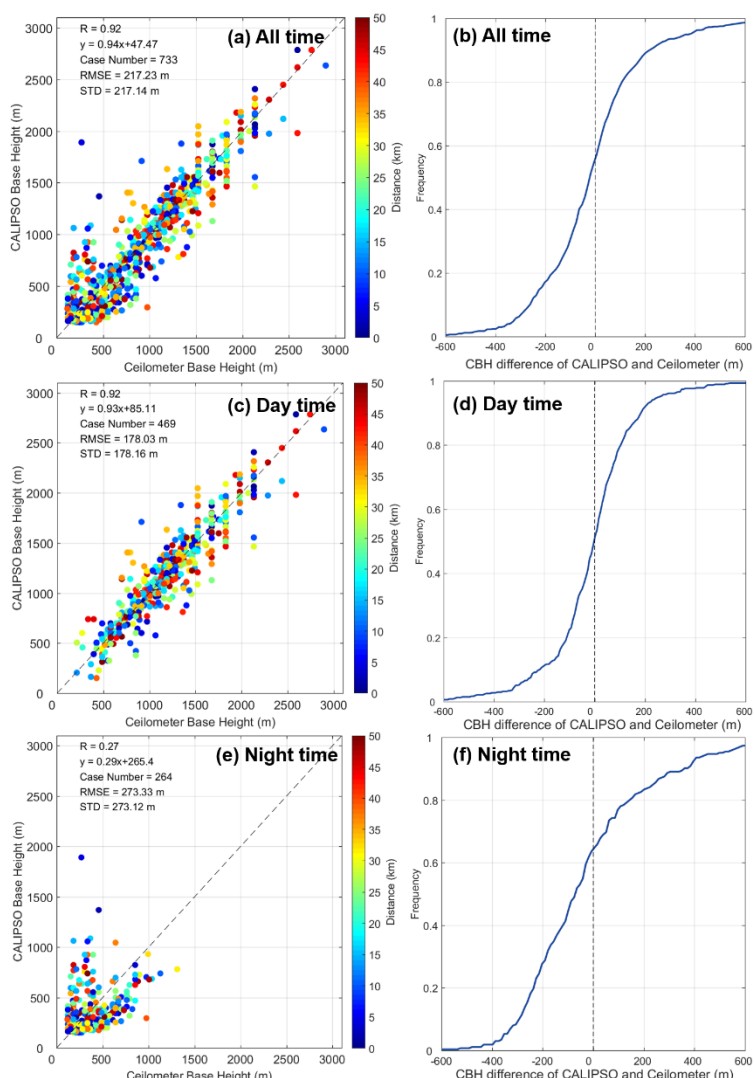

**Figure 8. Validation of CALIPSO-retrieved CBH against 138 continental ceilometer sites in the southern Great Plains in 2017-2020.** (a) Scatter plot of CALIPSO CBH and ceilometer CBH on all-time. The color represents the shortest distance from the CALIPSO ground track to the ceilometer site. (b) Cumulative distribution of the difference between CALIPSO CBH and Ceilometer CBH on all-time. (c)/(e) and (d)/(f) are the same as (a) and (b), but for day-time and night-time, respectively.

## 5 Global distributions

### 5.1 Overall distributions

Based on the above retrieval methodology, we further obtained the global geographic distribution of 2-year mean CBH, CTH, and CGT on 2° × 2° latitude-longitude grids in 2014 and 2017 (Figure 9). The CALIPSO CBH retrieval domain over land is 50 km and over the ocean is 100 km along the CALIPSO track. To ensure the validity of the retrieval results, we only use the

data at a 2-degree grid when there are more than 20 valid scenes on this grid, among which the valid scenes indicate that we have retrieved the CBH, CTH, and CGT based on the 333-m resolution cloud data. The blanks in the geographic distribution are mainly due to the lack of valid scenes for a given grid. This is more frequent over land than over the ocean, because there are more scenes with cloud bases above 3 km or more cloud-free scenes (e.g. Sahara, Australia).

The distribution of CBH above ground level (Figure 9a) shows that over land, CBHs are higher than over the ocean. In the oceanic area, the cloud bases are higher in the mid-latitudes than at the equatorial regions and at high latitudes, which are in good agreement with Mülmenstädt et al. (2018). In addition, in the mid-latitudes, the lowest cloud bases are mostly concentrated in offshore areas (Böhm et al., 2019), which are mainly less than 400 m. Clouds with high CTHs occur mainly over the ocean at low and high latitudes and over the land area, with CTHs over 2,000 m, which is consistent with the CTH

result of Sun-Mack et al. (2014). In particular, there is a peak area of CTH in the Tibetan Plateau region, essentially greater than 2800 m, which is consistent with the conclusions obtained by Yang et al. (2020) based on high spatial resolution Himawari imager data. Similar to the distribution of the CBH, the lowest CTHs which are mainly ~1,000 m are also concentrated in offshore regions and the equatorial regions of the western hemisphere, in agreement with Zuidema et al. (2009). Thus, shallow clouds with small CGTs (<800 m), with a percentage of ~10 % of all low-level clouds, occur mainly over mid-latitude oceanic

regions and eastern margins of the subtropical oceans. These areas mainly include the west coast areas of South America, Africa, the United States, and Australia, and are equally high-incidence areas of stratocumulus clouds (Wood, 2012). These shallow cloud geometric data retrieved in this study will be helpful to future studies of marine stratocumulus microphysics and aerosol-cloud interaction. Thick clouds with large CGTs are mainly located in the tropics and the mountainous regions, such as the western Pacific and the Rocky Mountains of western Canada.


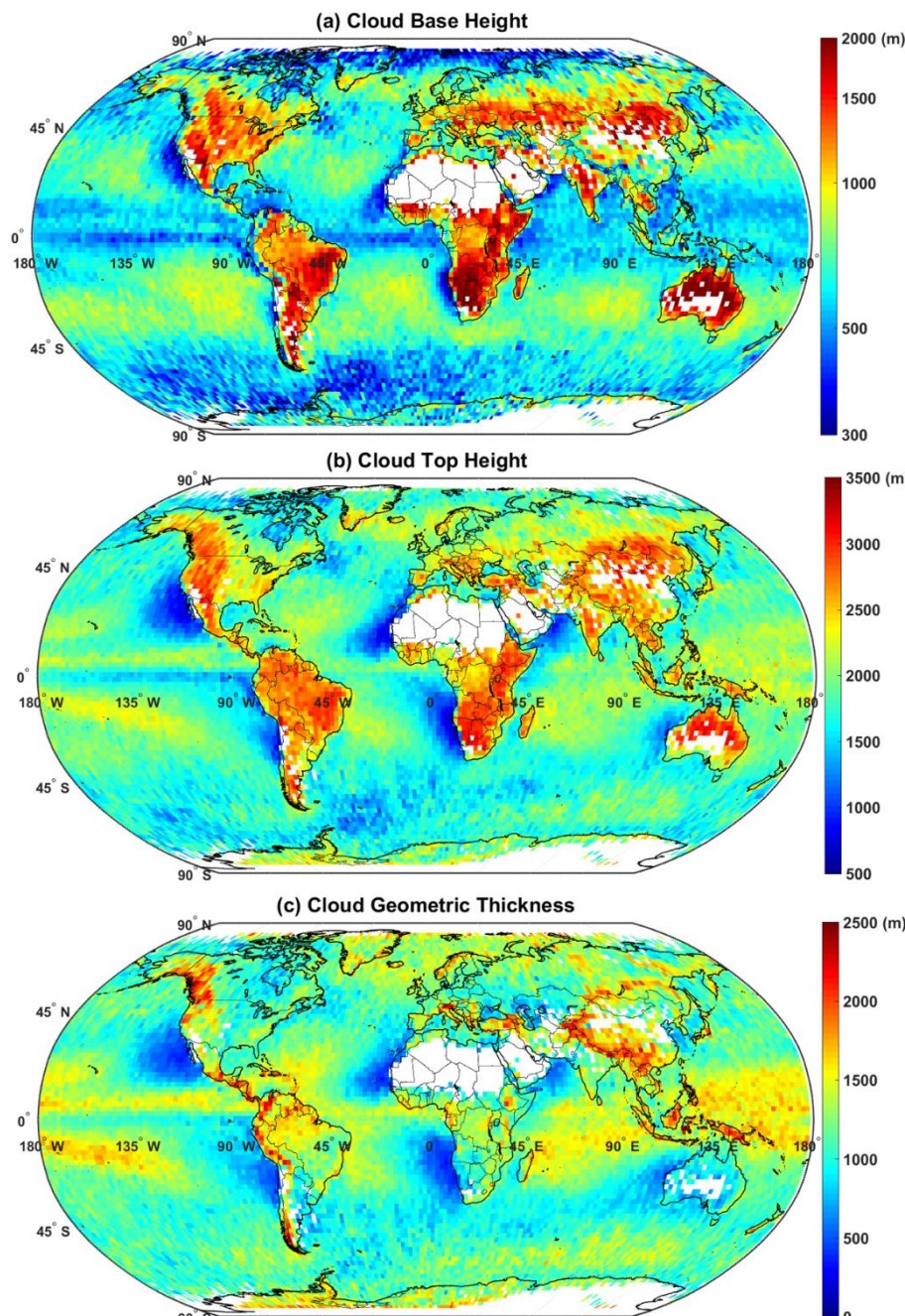

**Figure 9: Geographic distributions of 2-year mean CBH, CTH, and CGT on a 2° × 2° latitude-longitude grid in 2014 and 2017.** The heights are in m above ground level.

In addition, we obtained the geographic distribution of CTH by taking the maximum value of $H_{max}$ as the CTH of each 1°

CALIPSO scene (as shown in Figure A5). The spatial distribution of CTH in Figure A5a is similar to the distribution of CTH by using the mean height of the highest 10 percentile of $H_{max}$ as the CTH (Figure 9b). However, the cumulative distribution of

the difference (Figure A5b) indicates that the CTH based on the maximum value of $H_{max}$ is larger than that based on the mean height of the highest 10 percentile of $H_{max}$, ~60 % of cases have difference less than 100 m. The maximum difference reached 300 m, with ~3 % greater than 200 m. Those areas with high differences are concentrated in the oceans at low latitudes, which are mainly convective clouds.

We also counted the ratio of scenes that were rejected based on each criterion (as shown in Figure A6 in the Appendix). The results show a global average rejection ratio of ~29.5 %, which is mainly influenced by penetration efficiency (penetration efficiency of 333-m resolution cloud: 28.4 %; penetration efficiency of all resolution cloud: 29.5 %), with less influence from multilayer clouds. In addition, the results in Figure A6a show that a higher rejection ratio is at high latitudes than at middle and low latitudes, particularly in the Southern Ocean region.

## 5.2 Seasonal distributions

The seasonally averaged geographic distributions of CBH, CTH, and CGT (Figures 10, 11, and 12) are generally consistent with the distributions of the annual averaged results (Figure 9), but are influenced by the variation of convective intensity in different seasons, and also exhibit some unique seasonal characteristics. The CBH and CTH over land in the mid-latitude Northern Hemisphere are much greater in June-July-August (JJA) than other seasons and lowest in December-January-February (DJF). The mid-latitude global oceans have the lowest CBH (mostly below 500 m), CTH and CGT in summer seasons, and the highest cloud top height in winter, while the CBH, CTH, and CGT distribution reversed over high-latitude Southern Oceans. Previous study has also shown the same seasonal pattern of CBH and CTH distributions (Böhm et al., 2019), but with minimum discernible CBH > ~700 m.

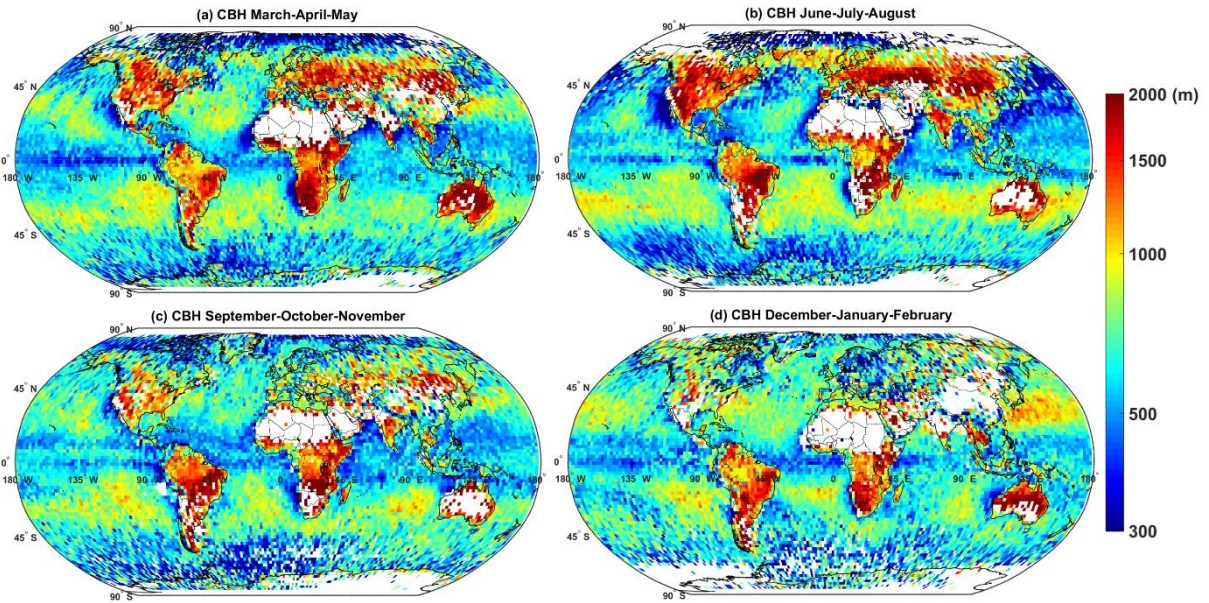

**Figure 10: Geographic distributions of seasonal averaged CBH on a 2° × 2° latitude-longitude grid in 2014 and 2017.** (a) March, April, and May; (b) June, July, and August; (c) September, October, and November; (d) December, January, and February. The heights are in m above ground level. Each grid has at least 5 valid scenes.

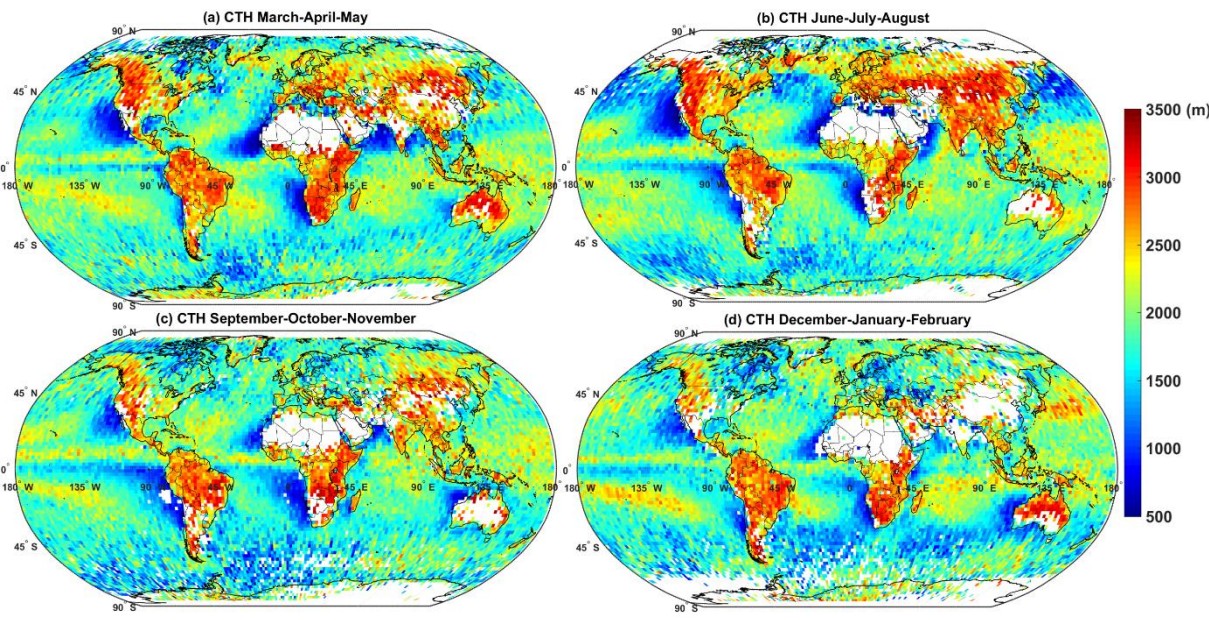


**Figure 11: The same as Figure 10, but for CTH.**

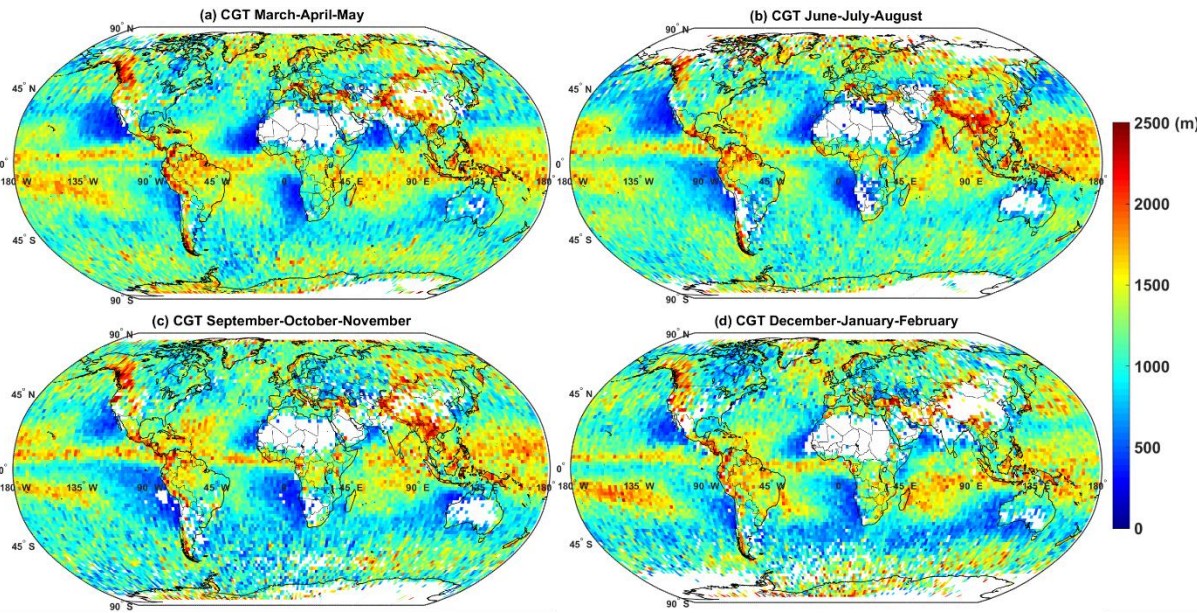

**Figure 12: The same as Figure 10, but for CGT.**

To further investigate the spatial variation of the cloud geometry information over land and ocean with different seasons, we

plotted the mean meridional distribution maps (Figure 13). Regional variations in CBH and CTH are more pronounced over

land than over the ocean during all seasons. The CTH follows CBH, especially over land. The seasonal variations are smaller over the ocean, and CBH and CTH show the maximum variations at the winter subtropical latitudes and the summer mid-latitudes, respectively. An equatorial minimum occurs in all seasons.

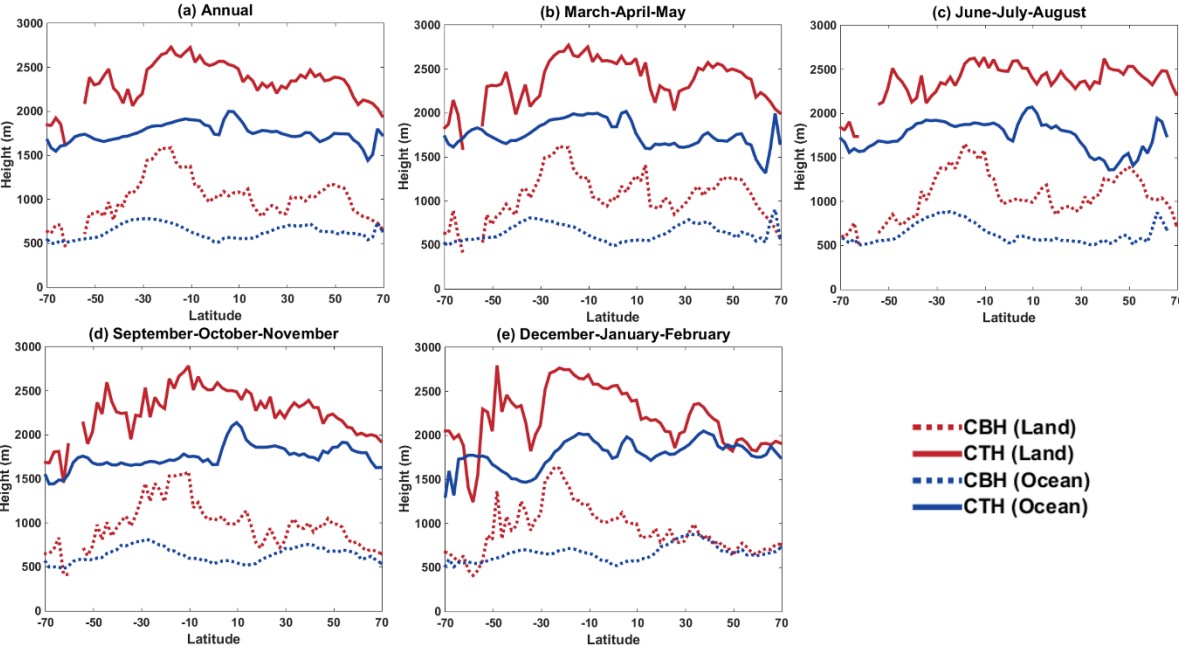

**Figure 13: Mean meridian CBH and CTH annual and seasonal distributions.** Straight lines indicate CTH, and dotted lines indicate CBH (ocean in blue, land in red). (a) Annual; (b) March, April, and May; (c) June, July, and August; (d) September, October, and November; and (e) December, January, and February.

## 5.3 Diurnal distributions

For the comparison of day-time and night-time distributions, we further obtained the diurnal geographical and mean meridional
distributions of CBH and CTH in Figure 14. Over land at mid and low latitudes, more boundary layer clouds are detected at day-time than at night-time, as evident by the sparse coverage on the geographic distribution maps. This means that either there are fewer clouds during the night, or the clouds become multilayer or obscured by deep or high clouds. The opposite is true for the Southern Oceans region. Overall, the CTHs over land are much higher during the day-time than at night-time, while over the ocean they are opposite, night-time has slightly higher CTH than day-time by ~300 m as shown by the
meridional mean distributions. The CBH over land is greater during day-time than at night-time, but over the ocean, the CBH is consistent both day-time and night-time. The difference of CTH and CBH between land and ocean is greater during day-time than at night-time, especially the CTH.

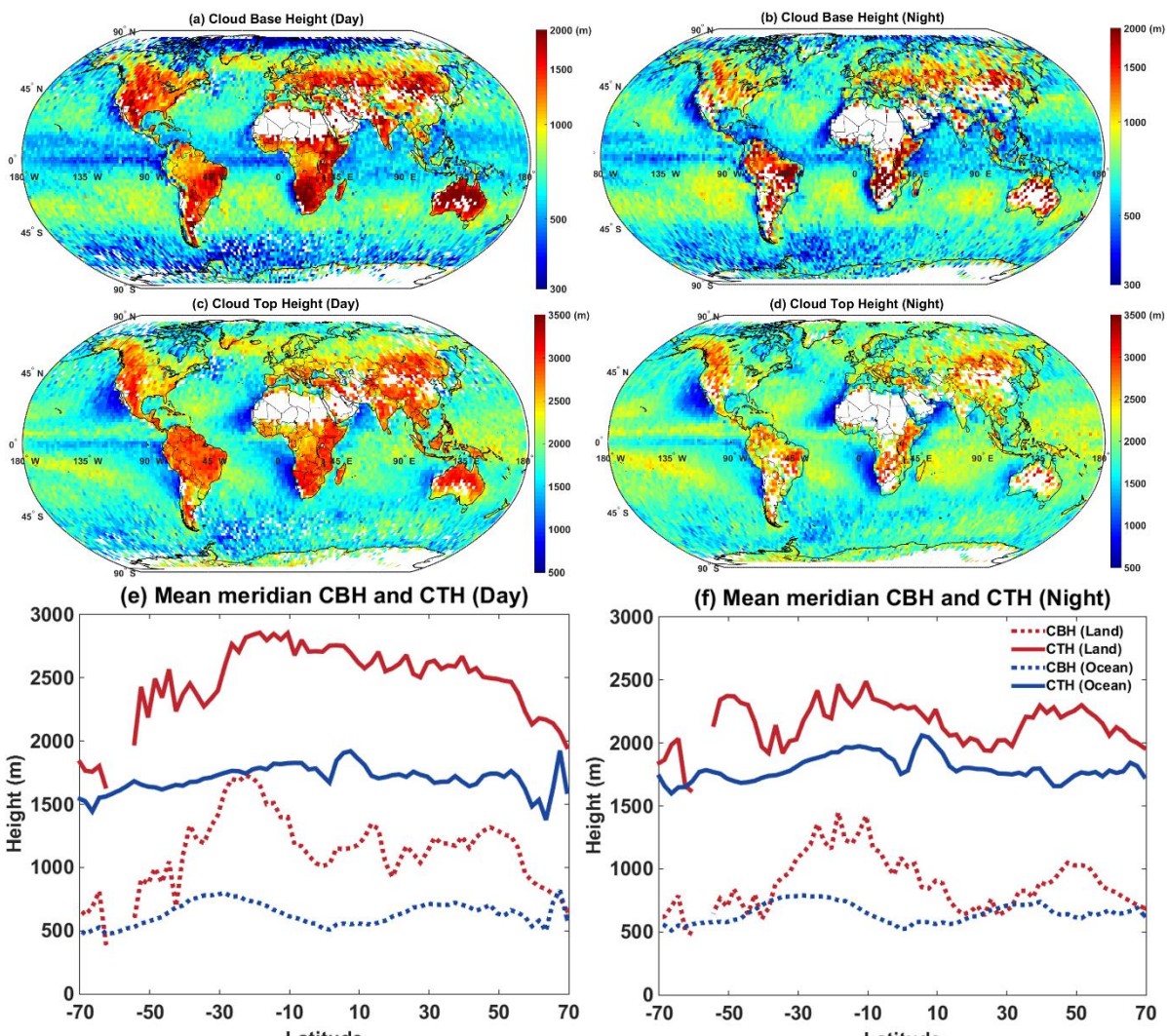

**Figure 14: Geographic distributions of annual mean CBH on a 2° × 2° latitude-longitude grid in 2014 and 2017 for** (a) CALIPSO day-time; (b) is the same as (a), but for night-time; (c) is the same as (a), but for CTH; (d) is the same as (c), but for night-time. The heights are in m above ground level. Each grid has at least 10 valid scenes. (e) Mean meridian CBH and CTH over land and the ocean for CALIPSO day-time. Straight lines indicate CTH, and dotted lines indicate CBH (ocean in blue, land in red). (f) is the same as (e), but for night-time.

## 6 Discussion

The CBH retrieval accuracy in this study has been greatly improved compared to other current satellite CBH retrieval algorithms. This allowed gaining new insights. According to the geographic distribution maps (Figure 9) generated from the high-accuracy CBH, CTH, and CGT data in this study, we find the following features that can be further discussed:

i. **The effect of SST on CBH and CGT.** The patterns show the obviously higher CBH over land compared to the ocean, especially over the arid areas. Shallow and thin clouds prevail over cool sea surface temperatures (SST) to the west of

subtropical coastlines, and thicken gradually westward. The CBH increases faster than CGT with the distance from land in these regions. The thickest clouds occur over the regions with the highest SST, such as the tropical west Pacific. A conspicuous narrow strip of clouds with low CBH and CGT is noted over the Equator, most notably over the eastern half of the Pacific Ocean. This feature is the manifestation of the equatorial ocean upwelling and cooling in response to the poleward flow of surface water in response to the easterly stress by the winds (Adam, 2020).

ii. **High base and thick clouds over tropical basins.** CBH is lowest near the shores of South America, Namibia, and Africa, because of the flow of warm continental air over the cold sea surface, which creates a strong inversion above it. This, in fact, leads to the formation of low cloud decks near the sea surface. This effect is weakening with distance from the shore. Therefore, these coastal regions are dominated by stratocumulus clouds, with CBH mainly below ~400 m and increasing up to ~1000 m far from the continent (the detailed distribution of CBH of the Southeast Atlantic is shown in Figure 15a), which is consistent with Andersen et al. (2019). The retrieved lower CBHs compared to the previous study (Mülmenstädt et al., 2018) make it possible to estimate the coupling state and its relevance to the effects of aerosols on cloud fraction based on this dataset. Over the tropical basins, such as the Amazon Basin and the Congo Basin, clouds developed high and thick with CBH larger than 1000 m and CGT larger than 1500 m (the detailed distribution of CBH of the Congo Basin is shown in Figure 15b), which responds to the strong convective motion in the tropics (Sun-Mack et al., 2014).

ii. **Large CGT and small CBH over the Southern Oceans.** CBH is quite low over the low SST of the Southern Oceans, but CGT is much larger there than over the eastern margins of the subtropical oceans. The lowering of CBH towards Antarctica in the Southern Oceans is caused by the more frequent and stronger thermal inversions at high latitudes (Li et al., 2013), with CBH largely below 500 m. This is much lower than the ~800 m CBH in the Southern Oceans inferred in previous studies (Böhm et al., 2019).

iii. **Low and thick clouds over the Maritime Continent.** CBH increases rapidly over the inland of tropical Africa, America and Australia, while keeping CGT little changed. The CBH increases much less over the Maritime Continent. Convection develops vigorously over those land areas, resulted in large CGT mainly greater than 1700 m, whereas the surrounding ocean area CGT<1700 m (the detailed distribution of CGT of the Maritime Continent is shown in Figure 15c).

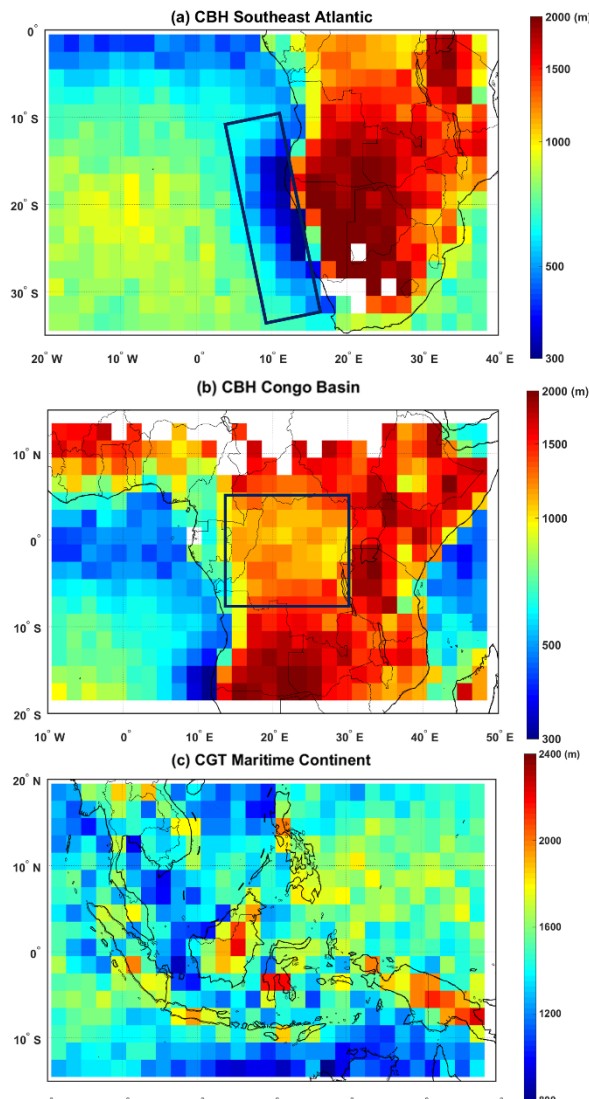

**Figure 15: Geographic distributions on a 2° × 2° latitude-longitude grid in 2014 and 2017**. (a) CBH over the Southeast Atlantic; (b) CBH over the Congo Basin; (c) CGT over the Maritime Continent. The heights are in m above ground level. The dark blue boxes represent the location of the regions of interest.

## 7 Conclusions

Based on the highest resolution VFM data of CALIPSO lidar observations, a new methodology for retrieving the CBH, CTH, and CGT of low-level clouds is proposed. This methodology uses the 333-m resolution water cloud distribution of VFM data to retrieve CBH with superior performance. The methodology can effectively reduce the interference to CBH retrieval due to surface signal, multilayer cloud, and boundary layer aerosols. Moreover, even when the thickness of the cloud is sufficient to

fully attenuate the CALIPSO lidar signal, the method provides an accurate CBH by taking the CBH of the surrounding thinner cloud as representative of the entire cloud field. At the same time, we used 1° scene along the CALIPSO track for CBH retrieval.

In addition, the 10 % percentile of all cloud base information, and the 40 % quantile of the first local peak was used as initial CBHs for over the ocean and land, respectively. All of these operations can reduce the effect of cloud anvils on the CALIPSO retrieval of CBH to some extent. The methodology was developed based on observations for the year 2017 from two ocean ceilometer stations. The Pearson correlation coefficient is 0.87, and an error standard deviation is ±115 m. Validation based on 4 years' data of 138 terrestrial ceilometer sites shows that the algorithm is applicable on land with R of 0.92 and an error

standard deviation of ±220 m. The land algorithm differs by when taking the 40 percentile of the first local peak of CALIPSO CBH, instead of the 10 percentiles of all cloud base heights over the ocean. This high-precision CBH retrieval methodology developed in this study is a great improvement over other current satellite CBH retrieval methods with RMSE of several hundred meters or even several km.

Based on this methodology, we obtained the annual, seasonal, and diurnal distributions of global CBH, CTH, and CGT for

two years. The lowest cloud base/top heights are both concentrated in the eastern margins of the oceans in the subtropical latitudes. A narrow band of lower clouds occurs along the Equator. Seasonal analysis showed that differences in CBH and CTH were more pronounced over land than over the ocean. The seasonal variation of CBH and CTH is greater in the Northern Hemisphere than in the Southern Hemisphere, both over land and over the ocean. The diurnal distribution suggests that CTH is much higher over land during the day-time than at night-time, while this phenomenon is mirrored and much weaker over

the ocean. This high-precision cloud geometry information also shows several interesting features: (1) there are noticeable differences in cloud geometry characteristics between the eastern and western parts of the Pacific Ocean; (2) high base and thick clouds occur over tropical basins; (3) the lowering of CBH towards Antarctica in the Southern Oceans, while deepening CGT; (4) low and thick clouds occur over the Maritime Continent.

Accurate CBH information is of great significance for evaluating the cloud coupling state and its relevance to the effects of

aerosols on cloud cover (Goren et al., 2018). The result in this study can also be applied to understanding the cloud microphysical processes and improve the accuracy of cloud radiation feedback in the numerical model (Hartmann, 2009;Jian et al., 2006;Merk et al., 2016;Viúdez-Mora et al., 2015). However, we can only retrieve CGT by leaving out the high multilayer cloud fraction and low penetration efficiency of CALIPSO VFM data. Therefore, the current method cannot deal with CALIPSO scenes with a large amount of multilayer clouds and non-penetration optical thick clouds.

**Appendix A**

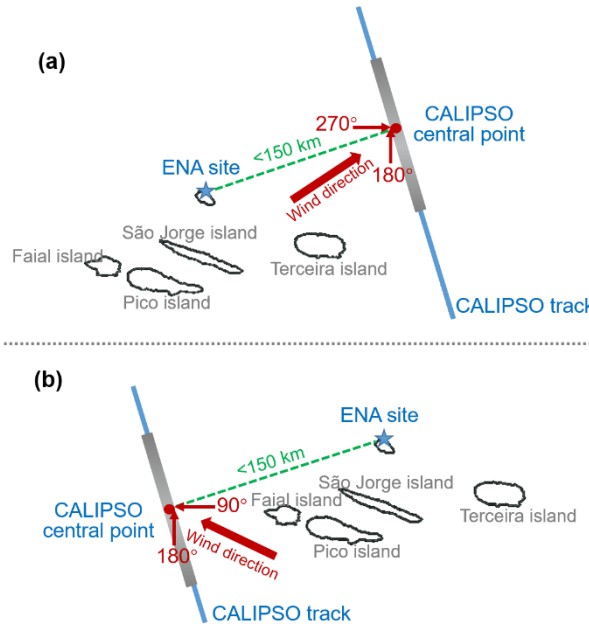

**Figure A1: Schematic of matching between the CALIPSO and the ENA observation site. The blue pentagram indicates the location of the ceilometer site, the blue line is the CALIPSO track, the red circle represents the CALIPSO central point, the dashed green line represents the shortest distance from the site to the CALIPSO central point, the grey shaded area is the 1° CALIPSO scene, and**
**the red arrow represents the wind direction of the CALIPSO data. (a): CALIPSO track is located at the east ocean of ENA site. Wind directions of rejected CALIPSO data: 180-270°. (b): CALIPSO track is located at the west ocean of ENA site. Wind directions of rejected CALIPSO data: 90-180°.**

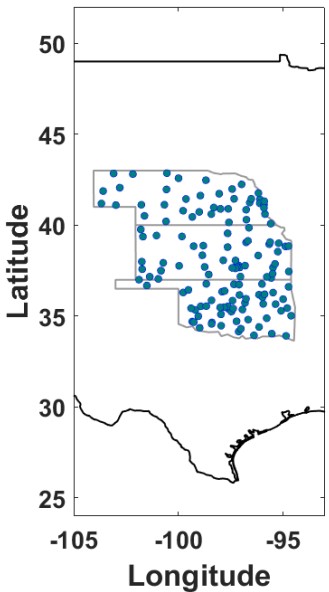

**Figure A2. The ASOS ceilometer distribution over land used for CBH evaluation.**

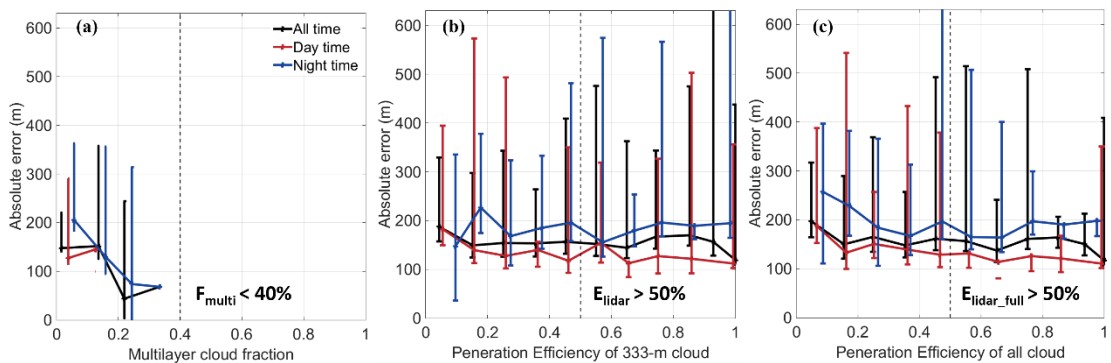


**Figure A3. (a) Joint distribution of CBH absolute error (between CALIPSO CBH and ceilometer CBH) and multilayer cloud fraction of 138 continental ceilometer sites in the southern Great Plains in 2017-2020 with distance less than 50 km. The different coloured line represents at different time (the black line: all time; the red line: day-time; the blue line: night-time). The I-beam line represents the standard error. (b) is the same as (a), but for penetration efficiency of 333-m resolution cloud. (c) is the same as (a), but for**
**penetration efficiency of all clouds.**

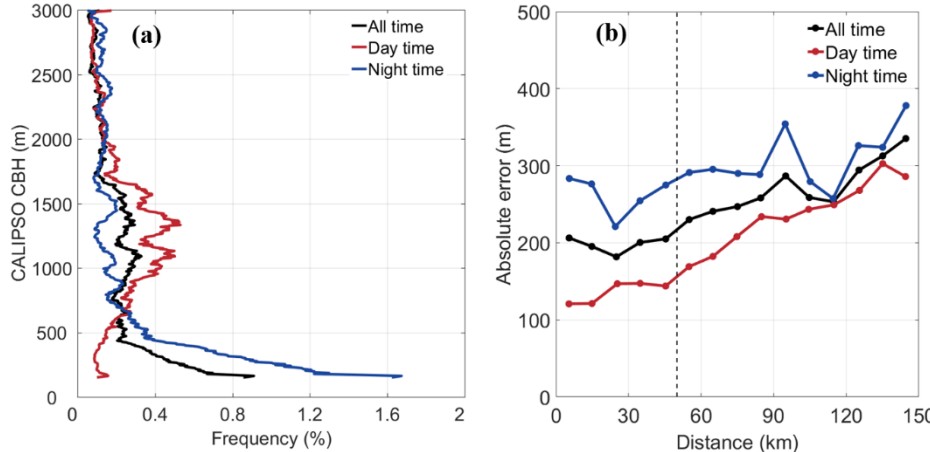

**Figure A4.** (a) Frequency profile of CALIPSO-retrieved CBH of 138 continental ceilometer sites in the southern Great Plains in 2017-2020. The different coloured line represents at different time (the black line: all time; the red line: day-time; the blue line: night-time). (b) Joint distribution of CBH absolute error (between CALIPSO CBH and ceilometer CBH) and distance (that is the shortest distance from the CALIPSO ground track to the ceilometer site) of 138 continental ceilometer sites in the southern Great Plains in 2017-2020. The different coloured line represents at different time (the black line: all time; the red line: day-time; the blue line: night-time).

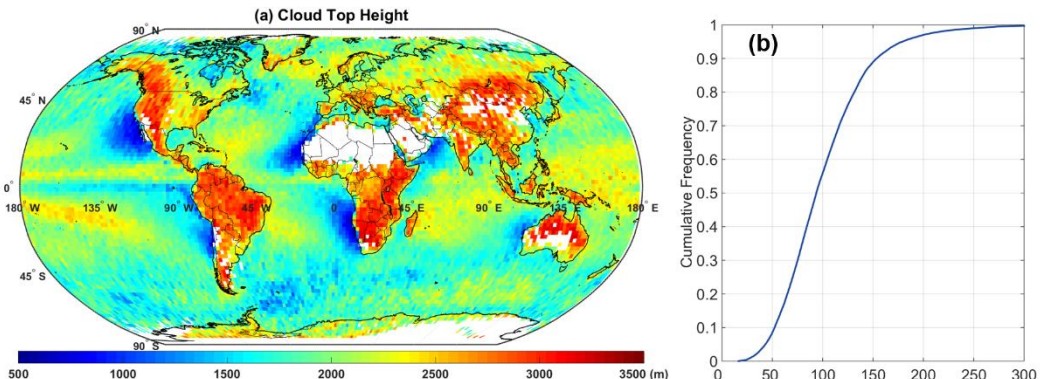

**Figure A5:** (a) Geographic distribution of CTH based on the maximum value of $H_{max}$ on a 2°× 2° latitude-longitude grid in 2014 and 2017. (b) Cumulative distribution of the difference between the CTH based on the maximum value of $H_{max}$ and CTH based on the mean height of the highest 10 percentile of $H_{max}$ in 2014 and 2017. The heights are in m above ground level.

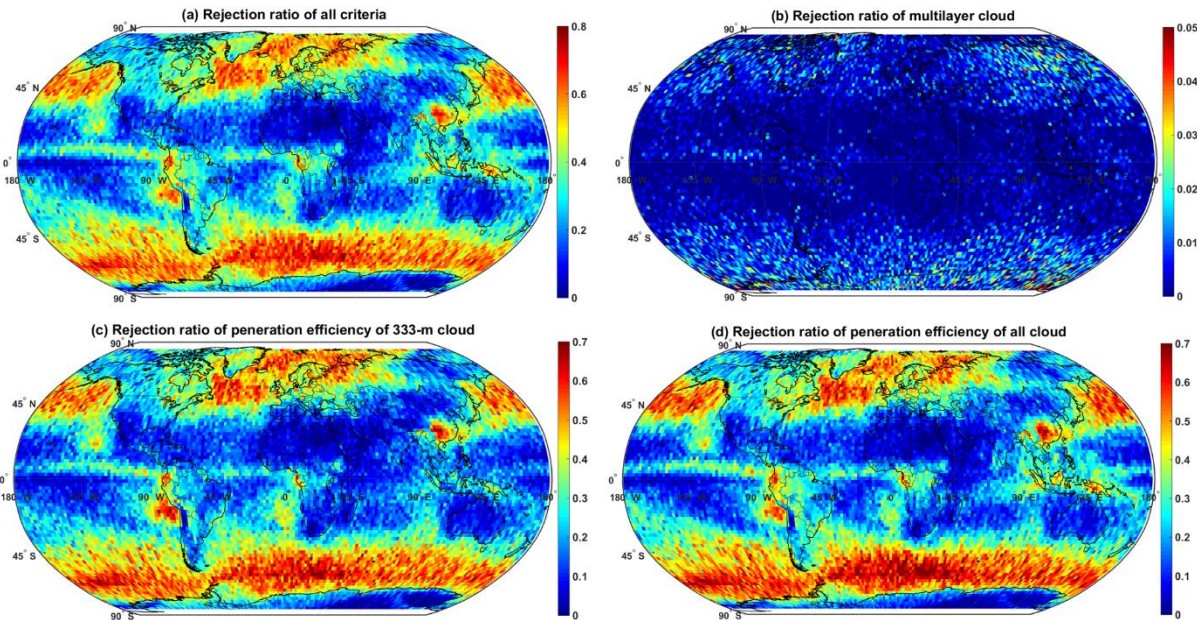

**Figure A6. (a) Geographic distribution of rejection ratio of all criteria (multilayer cloud, penetration efficiency of 333-m resolution cloud, and penetration efficiency of all resolution cloud) on a 2°× 2° latitude-longitude grid in 2014 and 2017. (b), (c) and (d) are the same as (a), but for multilayer cloud, penetration efficiency of 333-m resolution cloud, and penetration efficiency of all resolution cloud, respectively.**


**Appendix B**

**Table B1. Validation statistics of CALIPSO-retrieved initial CBH for different conditions of 138 continental ceilometer sites in the southern Great Plains in 2017-2020.**

|  | Case number | RMSE (m) |
|---|---|---|
| min CALIPSO CBH | 8404 | 635 |
| $H_{min}$ at 10 % quantile (same as ocean) | 7963 | 665 |
| 10 % quantile of the first local peak | 8302 | 616 |
| 20 % quantile of the first local peak | 8280 | 615 |
| 30 % quantile of the first local peak | 8270 | 619 |
| **40 % quantile of the first local peak** | **8203** | **612** |
| 50 % quantile of the first local peak | 8225 | 618 |
| 60 % quantile of the first local peak | 8193 | 619 |
| 70 % quantile of the first local peak | 8169 | 628 |



**Table B2. Final retrieval criteria used over the ocean and land.**

| Criteria | Over the ocean | Over land |
|---|---|---|
| Retrieval domain | <100 km | < 50 km |
| Initial CALIPSO CBH | 10 % percentile of all cloud base | 40 % quantile of the first local peak |
| $F_{multi}$ | <40 % | <40 % |
| $E_{lidar}$ | >50 % | >50 % |
| $E_{lidar\_full}$ | >50 % | >50 % |

**Data availability:** The CALIPSO VFM data used in this study can be downloaded from https://search.earthdata.nasa.gov/search/granules?p=C1556717896-LARC_ASDC&tl=1578270083!4 (last access: October 2020), Atmospheric Radiation Measurement ceilometer observation data are obtained from

https://adc.arm.gov/discovery/#/results/site_code::ena/meas_category_code::cloud/meas_subcategory_detail::cloud.macro (last access: October 2020), Barbados Cloud Observatory data are available at https://www.mpimet.mpg.de/en/science/the-atmosphere-in-the-earth-system/working-groups/tropical-cloud-observation/barbadosstation1/instrumentation-and-data/ (last access: October 2020) and ASOS ceilometer data source is https://mesonet.agron.iastate.edu/request/download.phtml (last access: May 2021).

**Author contribution:** DR conceived the study. XL, YZ, and DR designed the cloud base height and geometric thickness retrieval methodology. XL implemented the methodology and carried out the data analysis with help from YZ and FM. ZP, FM, and WG provided useful comments on the paper. XL prepared the manuscript with contributions from YZ, DR, and the other co-authors.

**Competing interests:** The authors declare that they have no conflict of interest.

**Acknowledgements:** This work was supported by the National Natural Science Foundation of China (41971285, 41627804, and 42075093), the National Key Research and Development Program of China (2018YFC1507903), and the Fundamental

Research Funds for the Central Universities (2042019kf0192). We are grateful to the science teams for providing excellent and accessible CALIPSO, Atmospheric Radiation Measurement ceilometer observation, and Barbados Cloud Observatory data.

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
