# Peer review of "Satellite retrieval of cloud base height and geometric thickness of lowlevel cloud based on CALIPSO"

_Atmospheric Chemistry and Physics, 2020_

## Author Comment (AC1)

Atmos. Chem. Phys. Discuss., referee comment RC1 https://doi.org/10.5194/acp-2020-1252-RC1, 2021

[Figure]

**Comment on acp-2020-1252**

**Anonymous Referee #1**

Referee comment on "Satellite retrieval of cloud base height and geometric thickness of low-level cloud based on CALIPSO" by Xin Lu et al., Atmos. Chem. Phys. Discuss., https://doi.org/10.5194/acp-2020-1252-RC1, 2021

**General comments**

Lu et al. present a novel algorithm to infer cloud base height from satellite-based observations, namely from CALIOP on CALIPSO. This is motivated because existing approaches suffer from high uncertainties which complicate their applicability for certain applications, such as studying cloud-aerosol interactions, determining cloud subadiabaticity, downwelling long wave radiation at the surface, etc. Therefore, the manuscript addresses an important issue.

The VFM, an operational product derived from CALIOP observations, is used as starting point for the CBH retrieval. For a cloud scene which comprises a distance of about 100 km, vertical profiles are filtered according to some requirements and the minimum heights ($H_{min}$) of liquid water cloud features are extracted. By taking the $10^{th}$ percentile of $H_{min}$, the CBH for is determined for this distance. The algorithm is only applied if further conditions are fulfilled regarding the fraction of multi-layer clouds, cloud cover fraction and detection efficiency for each scene. The percentile, along with these scene characteristics are evaluated according to a comparison with ground-based CBH measurements. Ceilometer observations from two island stations in the North Atlantic are utilized as ground-based reference.

**Question 1:**

The idea for this new approach seems to be inspired by Mülmenstädt et al. (2018) and Böhm et al. (2019). However, the authors should state this more clearly, addressing the similarities. In particular, the algorithm derived by Mülmenstädt et al. (2018) should be described in more detail with the goal to clearly distinguish the proposed algorithm from the former one and to state where the idea comes from and what has been adopted.

The main difference to the approach by Mülmenstädt et al. appears to be the way the cloud base height is derived for a specific point/area. Mülmenstädt et al. take the mean of all considered VFM CBH retrievals within a distance of 100 km weighted by estimated uncertainties to determine the CBH at a given point of interest. Lu et al. use the $10^{th}$ percentile of all retrievals within a similar area to calculate the CBH representative for the whole area. How these differences might lead to a better agreement with the reference data should be discussed in detail. Another difference between these two approaches is the validation. While Mülmenstädt et al. use METAR reports (ceilometer heights)

from about 1500 stations across the continental USA, Lu et al. use two maritime sites (islands in the North Atlantic).

**RE:** we have added a detailed description of the algorithm of Mülmenstädt et al. (2018) and explicitly point out the differences between the algorithm of this study and theirs.

**Paragraph 4 in Introduction:**

Mülmenstädt et al. (2018) retrieved the global CBH using CALIPSO Vertical Feature Mask (VFM) data and evaluated the retrieval algorithm based on ground-based ceilometer observation from about 1500 stations across the continental USA. They extrapolated CBH information from a surrounding field onto profiles for which the lidar signal was attenuated using CALIPSO's VFM, and took the mean of all considered VFM CBH retrievals within a distance of 100 km weighted by estimated uncertainties to determine the CBH at a given point of interest, but their overall RMSE of CBH exceeded 500 m. This provided the basic idea and motivation to retrieve the CBH at a higher precision in this study. This basic idea is that the CBH of the optically thin clouds can be used as the CBH for the whole scene at a given range (~100 km).

**Paragraph 2 in Section 3.1:**

Thus, we chose to use 333-m high resolution instead of other resolutions for CBH and CGT retrieval. This is one of the main differences between our algorithm and other current algorithms, such as Mülmenstädt et al. (2018). This largely reduces the impact on CBH retrieval due to official algorithmic misclassification of aerosols and clouds in low-resolution CALIPSO VFM data.

**Paragraph 2 in Section 3.2:**

Using $H_{min}$ at 10 % quantile as the initial CBH, rather than simply calculating the average CBH of $H_{min}$ as the initial value, goes some way to reducing the effect of high-altitude spreading of water clouds due to convective activity on the retrieval of CBHs.

**Paragraph 1 in Conclusions:**

At the same time, we used 1° scene along the CALIPSO track for CBH retrieval. In addition, the 10 % percentile of all cloud base information, and the 40 % quantile of the first local peak was used as initial CBHs for over the ocean and land, respectively. All of these operations can reduce the effect of cloud anvils on the CALIPSO retrieval of CBH to some extent.

Moreover, in addition to the currently available validation based on two maritime sites, we have added the additional validation experiments using 4 years (2017-2020) of ceilometer data from 138 continental sites in the Great Plains of the USA. Please see the below part of 'Regarding 1) Representativeness' for detailed responses.

**Question 2:**

The manuscript is generally well written and nicely structured. However, more details regarding the utilized data should be provided as indicated in the specific comments. Furthermore, I see three major issues with the provided study.

- Only two maritime reference sites are not convincing proof for the applicability of the algorithm on a global scale.
- The development of the algorithm should be improved.
- The validation (Section 4) should be carried out for an independent data set which has not been applied already to develop the algorithm.

**RE:** Thank you for your comments.

**1)** We have added the additional validation experiments using 4 years of ceilometer data from 138 continental sites in the southern Great Plains. Please see the below part of '*Regarding 1) Representativeness*' for detailed responses.

**2)** We have made improvements to the algorithm development and evaluation with reference to your suggestion. Please see the below part of '*Regarding 2) Algorithm development*' for detailed responses.

**3)** We have applied the CBH retrieval algorithm developed based on the cloud observations over the ocean to land (as an independent data set which has not been applied already to develop the algorithm) and obtained similar conclusions. Please see the below part of '*Regarding 1) Representativeness*' for detailed responses.

**Regarding 1) Representativeness**

The validation is only carried out over maritime regions and includes very few coincident satellite- and ground-based observations (after filtering 72 events remain). Mülmenstädt et al. (2018) argue that various cloud morphologies are represented in their continental validation data set but maritime stratocumulus might be underrepresented hinting that their algorithm might have trouble for this cloud type. Analogous, the problem for this study is the opposite featuring mainly maritime cumulus and stratocumulus clouds. From Figure 8 it becomes clear that the algorithm was developed and validated based on clouds with CBH mostly between 400 and 800 m. Therefore, before it can be applied outside this range, in particular over land where apparently higher CBHs dominate, a thorough discussion should be provided on the representativeness of the validation. This would necessitate including continental sites as ground-based reference.

**RE:** To validate the applicability over land of the retrieval algorithm in this study and to increase the case number for validation, we conducted additional validation experiments using 4 years of ceilometer data from 138 continental sites in the Great Plains of the USA (Figure R1, this figure is also provided as Figure A2 in the Appendix of the paper). These ceilometer data are derived from Automated Surface Observation System (ASOS). The data source is https://mesonet.agron.iastate.edu/request/download.phtml (last access: May 2021), which is maintained by Iowa Environmental Mesonet of Iowa State University. The data period used in this study is taken from 2017 to 2020 because during this period, the ceilometers provide better time resolution of cloud base measurements.

[Figure]

**Figure R1. The ASOS ceilometer distribution over land used for CBH evaluation.**

The ASOS uses a laser beam ceilometer with a time interval of 5-30 minutes, with a vertical resolution of ~30 m, and a vertical detection range of ~3700 m. In order to ensure the data quality of the ASOS ceilometer observations, we only use ceilometer data with CBH less than 3000 m and limit the number of valid cloud observations to no less than 2 during 1 hour.

Then, based on the CBH retrieval algorithm developed from the cloud observations over ocean, we conducted the same experiment on land to test the applicability of our algorithm. Since the cloud base is not as homogeneous over land as over ocean, we consider using the cloud information below the first peak nearest to the surface in the cloud fraction profile of 1° scene as a proxy for all cloud base information in this scene (which we defined as the first local peak above surface). In this way, we avoid missing the newly developed clouds with small size. Therefore instead of using $H_{min}$ at 10 % quantile of all clouds as the initial CBH over ocean, we tested the CBH at different quartiles of the first local peak as the initial CBH for the scene. Table R1 shows the corresponding statistical results (this table is also provided as Table B1 in the Appendix of the paper), which can be seen there is a minimum RMSE when the initial CALIPSO CBH is at the 40% quantile of the first local peak which closest to surface. This has also been verified in the results of other years (2007-2016, not shown). Thus on land the CALIPSO-retrieved initial CBH is 40% quantile of first local peak, while over the ocean it is $H_{min}$ at 10 % quantile.

**Table R1.** Validation statistics of CALIPSO-retrieved initial CBH for different conditions of 138 continental ceilometer sites in the southern Great Plains in 2017-2020.

|  | Case number | RMSE (m) |
| --- | --- | --- |
| min CALIPSO CBH | 8404 | 635 |
| $H_{min}$ at 10 % quantile (same as ocean) | 7963 | 665 |
| 10 % quantile of the first local peak | 8302 | 616 |
| 20 % quantile of the first local peak | 8280 | 615 |
| 30 % quantile of the first local peak | 8270 | 619 |
| **40 % quantile of the first local peak** | **8203** | **612** |
| 50 % quantile of the first local peak | 8225 | 618 |
| 60 % quantile of the first local peak | 8193 | 619 |
| 70 % quantile of the first local peak | 8169 | 628 |

Then, based on the CBH data obtained from the above processing, we further tested the effects of $F_{multi}$, $E_{lidar}$ and $F_{lidar\_full}$ over land following the same process as over ocean, as showed in Figure R2 (this figure is also provided as Figure A3 in the Appendix of the manuscript). From the results it can be seen that the optimal thresholds for $E_{lidar}>50\%$, and $F_{lidar\_full}>50\%$ over land are consistent with those over ocean, which also shows that the CBH retrieval algorithm we developed based on cloud observations from the ocean is applicable on land. The parameters $F_{multi}<40\%$ has little meaning over land due to the scarcity of the cases that fulfil that condition. It is kept over land for consistency with the ocean criteria.

[Figure]

**Figure R2. (a)** Joint distribution of CBH absolute error (between CALIPSO CBH and ceilometer CBH) and multilayer cloud fraction of 138 continental ceilometer sites in the southern Great Plains in 2017-2020 with distance less than 50 km. The different coloured line represents at different time (the black line: all time; the red line: day-time; the blue line: night-time). The I-beam line represents the standard error. **(b)** is the same as **(a)**, but for penetration efficiency of 333-m resolution cloud. **(c)** is the same as **(a)**, but for penetration efficiency of all cloud.

As mentioned before due to the complexity of topography and land surface situation, the cloud bases height varies at larger spatial scales. The 150 km distance between the shortest distance from the CALIPSO ground track to the ceilometer site cannot be used for over land validation. We have to shrink the distance to minimize the spatial variability due to the changes over land. We tested the effect of distance (that is the shortest distance from the CALIPSO ground track to the ceilometer site) and observation time on the retrieval results (Figure R3, which is also provided as Figure A4 in the Appendix of the manuscript). The results (Figure R3b) show that the absolute error between the CALIPSO CBH and the ceilometer CBH becomes smaller as the distance decreases, stabilizing at distances less than 50 km. It is therefore preferable to limit the distance to 50 km for studies on land to better meet the assumptions of a homogeneous CBH within 1° scenes. It can also be seen that the cloud base heights are more evenly distributed during the day-time (Figure R3a, 300-1800 m) than at night, while at night CBHs are mainly concentrated below about 700 m and are most frequent very near the surface, where validation becomes unreliable. In addition to the distance limitation, the cloud base homogeneity further constrained by comparing the lifted condensation level ($H_{LCL}$) to ceilometer cloud base height. To satisfy the cloud base homogeneity assumption (Efraim et al., 2020), cases are selected when the absolute difference between the $H_{LCL}$(calculated from ASOS-observed air temperature and dew point temperature) and ceilometer CBH is less than 200 m. In summary, the ceilometer measurements need to satisfy the following conditions for validating CALIPSO CBH retrieval: 1) The ceilometer is within 50 km radius to the center of CALIPSO ground track; 2) the ceilometer measured CBH should have an absolute difference less than 200 m against $H_{LCL}$ as calculated from surface measured air temperature and dew point.

[revised manuscript text omitted]

**Regarding 2) Algorithm development**

**Question 1:**

So far, the algorithm development (Section 3) comprises mainly the estimation of the following parameters: a suitable quantile to apply to the $H_{min}$ distribution; thresholds for $F_{multi}$, $F_{cloud}$, $E_{lidar}$. Furthermore, the VFM quality assurance (QA) flag could be considered (see Mülmenstädt et al., 2018).

**RE:** Thank you for your suggestion. In our algorithm, we use the 333-m high-resolution VFM data for the retrieval of cloud base heights. Retrieval of cloud base heights based on high-resolution VFM data is less susceptible to interference from boundary layer aerosols and surface echoes (Mace and Zhang, 2014;Vaughan et al., 2005). This is similar to the algorithm of Mülmenstädt et al. in which the use of high QA flag VFM data gives the highest accuracy in retrieval of CBHs. Moreover, the distribution of high QA 333-m resolution water clouds is shown in Figure R5c. We can see that the distribution of high QA 333-m resolution water clouds (Figure R5c) is similar to the distribution of 333-m resolution water clouds (Figure R5b). Some of the differences are circled in red in Figure R5. To ensure that high quality CALIPSO VFM data is used, we have currently updated the data and all figures used in the paper to high QA 333-m resolution water cloud data, and the results obtained remain largely unchanged from the previous results using only 333-m resolution water cloud data. We have also added the corresponding descriptions in the manuscript:

**Section 2.1:**

To ensure that high quality CALIPSO VFM data is used, we limit the VFM quality assurance flag as "high" (Mülmenstädt et al., 2018).

[Figure]

**Figure R5**. **Schematic of CALIPSO low-level cloud feature determination (CALIPSO VFM data at (UTC) 05:51:17 on 3 January 2017).** **(a)** Latitude-altitude distribution of feature type based on CALIPSO VFM feature type parameter, the light blue areas represent the water cloud features. "Strat" is the stratospheric feature, "SubSurf" is the Sub Surface. **(b)** 333-m horizontal resolution water cloud distribution; the light blue areas indicate the 333-m horizontal resolution water clouds, the green areas refer to aerosols at any resolution, the orange area is the surface. **(c)** Distribution of 333-m horizontal resolution water cloud with high quality assurance flag; the light blue areas indicate the high quality assurance water clouds, the green areas refer to aerosols at any resolution, the orange area is the surface.

**Question 2:**

Instead of testing out some example values, a more continuous approach would yield stronger arguments for the selected choices for these parameters. This could also be visualized as follows: Plotting Pearson correlation coeff., RMSE, Bias and maybe also the sample size in dependence on the parameters mentioned above ($H_{min}$, $F_{multi}$, $F_{cloud}$, $E_{lidar}$). This might be more informative than the scatter plots from Figures 4-6. In order to quantify uncertainties (add a confidence interval) for these

statistical measures (r, RMSE, Bias) a bootstrap analysis could be carried out (just an idea).

**RE:** Thank you for your good idea. Since there is less matching data on the ocean, we have maintained the previous presentation, but over land we use a joint distribution map to continuously show the effects of different parameters ($H_{min}$, $F_{multi}$, $E_{lidar}$ and $E_{lidar\_full}$). Please see the above part of '*Regarding 1) Representativeness*' for detailed responses (Figure R2, Figure R3 and Figure R4).

**Question 3:**

Furthermore, the overall r, RMSE and Bias should also be considered in dependence on the distance to the ceilometer (-> get a sense of the assumption of a homogeneous CBH field), some other cloud scene characteristics, such as number of valid VFM CBH retrievals within scene, CGT, CBH and/or CTH (get a sense if the error depends on the cloud scene). A distinction by day and night would also be beneficial. Depending on the nature of the bias, a bias corrected RMSE might be more appropriate.

**RE:** Thank you for your good idea. We have displayed the effect of distance and observation time (day and night) on the retrieval results in above '*Regarding 1) Representativeness*' and also added the appropriate content to the manuscript. Moreover, we further analysed the effect of distance and cloud scene characteristics (CALIPSO CBH, CALIPSO CGT, number of valid VFM CBH retrievals within 1° scene (Number_ValidCBH)) on the overall R, RMSE and STD based on day-time observations (as shown in Table R2). Statistics concerning cloud scene characteristics were all performed at distances less than 50 km. The results show that over land RMSE is ~310 m for distances between 50 km and 100 km, which is worse than for distances less than 50 km, but still better than most current other CBH retrieval algorithms. The retrieval results do not depend much on the height of the cloud base, but the RMSE increases as the CGT increases, with a minimum RMSE of ~140 m and a maximum RMSE of ~250 m. In addition, the retrieval results become better as the number of valid VFM CBH retrievals within scene increases, but the RMSE is below ~200 m for different numbers of valid VFM CBH retrievals.

**Table R2.** Statistics of validation results under different conditions over land in 2017-2020 on day-time. Number_ValidCBH is the number of valid VFM CBH retrievals within 1° scenes. The statistics concerning cloud scene characteristics (CALIPSO CBH, CALIPSO CGT and Number_ValidCBH) were all performed at distances less than 50 km.

|  | R | RMSE (m) | STD (m) |
|---|---|---|---|
| Distance<50 km | 0.92 | 178.0 | 178.2 |
| 50 km≤Distance<100 km | 0.79 | 313.2 | 312.4 |
| 100 km≤Distance<150 km | 0.75 | 366.8 | 356.6 |
| CALIPSO CBH<1000 m | 0.77 | 158.2 | 152.1 |
| 1000 m≤CALIPSO CBH<2000 m | 0.79 | 187.1 | 187.1 |
| 2000 m≤CALIPSO CBH<3000 m | 0.81 | 188.7 | 178.2 |
| CALIPSO CGT<1000 m | 0.95 | 136.5 | 134.7 |
| 1000 m≤CALIPSO CGT<2000 m | 0.95 | 163.7 | 163.8 |
| 2000 m≤CALIPSO CGT<3000 m | 0.92 | 190.0 | 194.8 |
| CALIPSO CGT>3000 m | 0.83 | 248.6 | 249.7 |
| Number_ValidCBH<50 | 0.91 | 209.7 | 210.5 |
| 50≤Number_ValidCBH<100 | 0.91 | 183.6 | 184.0 |
| 100≤Number_ValidCBH<150 | 0.93 | 168.3 | 166.0 |
| Number_ValidCBH>150 | 0.95 | 121.8 | 119.0 |

**Question 4:**

Some quantification of the algorithm efficiency would be desirable which quantifies how often a

successful retrieval is possible compared to the total number of cloudy scenes. How does the total number of CALIPSO overpasses split into cloud free scenes, valid retrieval, or neglected by criteria? Additionally, reasons for failure could be distinguished. Is there a distinct distribution globally, where the algorithm works effectively or not?

330 **RE:** Thank you for your suggestion. For all CALIPSO overpasses data, we only retrieve the cloud geometry information for scenes with cloud base heights below 3 km. These invertible scenes are considered valid if they meet the threshold conditions proposed above (multilayer cloud, penetration efficiency of 333-m resolution cloud, and penetration efficiency of all resolution cloud), otherwise they are rejected. We have counted the ratio of scenes were rejected based on which criterion (as 335 shown in Figure R6, which is also provided as Figure A6 in the Appendix of the manuscript).

The results in Figure R6 show a global average rejection ratio of ~ 29.5%, which is mainly influenced from penetration efficiency (penetration efficiency of 333-m resolution cloud: 28.4%; penetration efficiency of all resolution cloud: 29.5%), with less influence from multilayer clouds. In addition, the results in Figure R6a show that a higher rejection ratio are at high latitudes than at middle and 340 low latitudes, particularly in the Southern Ocean region. We have also added these information to the manuscript:

**Paragraph 4 in Section 5.1:**

We also counted the ratio of scenes were rejected based on each criterion (as shown in Figure A6 in the Appendix). The results show a global average rejection ratio of ~29.5 %, which is mainly 345 influenced from penetration efficiency (penetration efficiency of 333-m resolution cloud: 28.4 %; penetration efficiency of all resolution cloud: 29.5 %), with less influence from multilayer clouds. In addition, the results in Figure A6a show that a higher rejection ratio is at high latitudes than at middle and low latitudes, particularly in the Southern Ocean region.

[Figure]

350 **Figure R6**. **(a)** Geographic distribution of rejection ratio of all criteria (multilayer cloud, penetration efficiency of 333-m resolution cloud, and penetration efficiency of all resolution cloud) on a $2° \times 2°$ latitude–longitude grid in 2014 and 2017. **(b)**, **(c)** and **(d)** are the same as **(a)**, but for multilayer cloud, penetration efficiency of 333-m resolution cloud, and penetration efficiency of all resolution cloud, respectively.

*Specific comments*

1. Line 15: Should be clarified whether the sentence refers to the author's study, or is meant generally.
   **RE:** This sentence has been revised as "Satellite-based cloud base and top height (CBH and CTH) and cloud geometrical thickness (CGT) are validated against ground based lidar measurements and provide new scientific insights. The satellite measurements are done by the Cloud Aerosol Lidar and Infrared Pathfinder Satellite Observation (CALIPSO)" in Abstract.

2. Line 47: Could also cite Merk et al. 2016 (https://doi.org/10.5194/acp-16-933-2016) regarding adiabatic assumption.
   **RE:** We have added the citation of Merk et al. 2016 at Paragraph 1 in Introduction.

3. Line 79: Mülmenstädt et al. 2018 extrapolated CBH information from a surrounding field onto profiles for which the LIDAR signal was attenuated using CALIOP's VFM. It should be clearly stated that their study constitutes the basic idea for this study. Therefore, their retrieval approach should be explained in a bit more detail.
   **RE:** This description has been revised as "Mülmenstädt et al. (2018) retrieved the global CBH using CALIPSO Vertical Feature Mask (VFM) data and evaluated the retrieval algorithm based on ground-based ceilometer observation from about 1500 stations across the continental USA. They extrapolated CBH information from a surrounding field onto profiles for which the lidar signal was attenuated using CALIPSO's VFM, and took the mean of all considered VFM CBH retrievals within a distance of 100 km weighted by estimated uncertainties to determine the CBH at a given point of interest, but their overall RMSE of CBH exceeded 500 m. This provided the basic idea and motivation to retrieve the CBH at a higher precision in this study. This basic idea is that the CBH of the optically thin clouds can be used as the CBH for the whole scene at a given range (~100 km)" at Paragraph 4 in Introduction.

4. Section 2.1. More information on the satellite data, such as orbit characteristics, equatorial overpass time, should be provided.
   **RE:** The other detailed information of CALIPSO has been added as "CALIPSO, jointly developed by NASA and CNES, is a sun-synchronous orbiting satellite with an orbital inclination of 98.2°, an orbital altitude of 705 km, a revisit period of 16 days, and an equatorial crossing time of approximately 13:30 local time" at Paragraph 1 in Section 2.1.

5. Section 2.2. Information regarding limitations of the ceilometer should be provided (e.g. detection range, temporal & vertical resolution)
   **RE:** The limitations of the ceilometer has been provided in Section 2.2 as follow:
   The temporal resolution of the ceilometer at Barbados site is 10 seconds, and the vertical resolution is 15 m.
   At the ENA site, the ceilometer has a temporal resolution of 16 seconds, a vertical resolution of 30 m and a maximum detection range of 7700 m.
   The ASOS uses a laser beam ceilometer with a time interval of 5-30 minutes, with a vertical resolution of ~30 m, and a vertical detection range of ~3700 m. In order to ensure the data quality of the ASOS ceilometer observations, we only use ceilometer data with cloud base heights less than 3000 m.

6. I suggest to explain in more detail how the reference CBH based on the ceilometer measurements is derived. This should be done in this section (2.2.). It is mentioned in Section 3.3. that the 10 percentile of $CB_{ceilo}$ is taken as the "true" CBH. What is the rationale behind that?
   **RE:** Since averaged $CB_{ceilo}$ can be influenced by the developed high-level clouds during the observation period and the minimum of $CB_{ceilo}$ can be influenced by the surrounding surface undulations, we used the 10 percentile of $CB_{ceilo}$ instead of the averaged or the minimum of

CB$_{ceilo}$ as the "true" CBH, which is also consistent with the treatment of CALIPSO data in this study.

The description of the extraction of the reference CBH based on the ceilometer measurements has also been moved to Section 2.3 (Data matching) as follow:

Then we obtained the distribution of the base height of cloud features observed by ceilometers within 30 minutes before and after the CALIPSO overpass time (CB$_{ceilo}$). The lowest 10 % quantile of the CB$_{ceilo}$ is determined as true CBH. The measurements avoid the possible influence of developing high-level clouds and surface undulations around the ceilometer site.

7. Line 114: It would be worth mentioning what type of clouds can be expected at these two validation sites. The paragraph should be split here (new topic: data matching).

   **RE:** The cloud type description has been added at Paragraph 1 in Section 2.2 as " The data period used in this study of these two marine sites is from January 2017 to December 2017, and the cloud types are mostly marine stratocumulus and trade wind cumulus".

   The paragraph also has been split here and we added a new Section 2.3 (Data matching).

8. Line 115: "a scene of 1° is selected" – Is it 1° along the perimeter of the Eath in along-track direction? Please, clarify what this refers to.

   **RE:** This sentence has been revised as "a scene of 1° along the CALIPSO track is selected…" in Section 2.3.

9. Section 3.1. Line 150: If a comparison with a retrieval based on the adiabatic assumption is included, more details should be provided on how this is carried out. However, this adiabatic assumption is rather uncertain as stated in the introduction, so this is a weak argument here. The authors could test how the horizontal resolution of the VFM influences the uncertainty of the derived CBH retrieval.

   **RE:** The description has been revised as "Moreover, CBH obtained from higher resolution VFM data was closer to the lifting condensation level (Seung-Hee et al., 2017)." at Paragraph 2 in Section 3.1.

10. Figure 2: (a) Label-color correspondence is unclear and should be improved. (b) Caption states that light blue represents 333m even though the colobar indicates gray color.

    **RE:** Sorry for that. The label-color and caption of Figure 2 have been revised as follow:

[Figure]

**Figure 1: Schematic of CALIPSO low-level cloud feature determination (CALIPSO VFM data at (UTC) 05:51:17 on 3 January 2017).** **(a)** Latitude-altitude distribution of feature type based on CALIPSO VFM feature type parameter, the light blue areas represent the water cloud features. "Strat" is the stratospheric feature, "SubSurf" is the Sub Surface. **(b)** Resolution information distribution based on CALIPSO VFM resolution parameter, the gray areas represent the 333-m resolution features. **(c)** 333-m horizontal resolution water cloud distribution combined by **(a)** and **(b)**; the light blue areas indicate the 333-m horizontal resolution water clouds, the green areas refer to aerosols at any resolution, the orange area is the surface.

11. Section 3.2, line 165: "we follow a main hypothesis" – I think, the authors mean "assumption" instead of "hypothesis".

   **RE:** This "a main hypothesis" has been revised as "a main assumption" at Paragraph 1 in Section

3.

12. Line 166: "which is the lifting condensation level of coupled clouds" – Remove. Repetition from above; also this assumption is applied generally. There is no filtering for coupled only clouds.

**RE:** This sentence has been removed in Section 3.2.

13. Line 167: If this assumption is valid, then the CBH from thin clouds may serve as a proxy. However, there is no proof of such validity. Citing Mülmenstädt et al. (2018) here might suggest that there is proof even though there is not.

**RE:** Sorry for that. As this section (Section 3.2) is similar to the first paragraph of Section 3, we have merged it and amended it accordingly as follows:

**Paragraph 1 in Section 3:**

That formative cloud base level is similar in areas with similar thermodynamic structure, which is conducive to a nearly constant cloud base height. Thus, the CBH of optically thicker clouds that cannot be penetrated by CALIOP can be expressed by the CBH of the surrounding penetrable thin clouds. The retrieval algorithm relies on this assumption, by adopting the lowest reliably detected cloud height along a CALIPSO track of approximately 100 km (1° along the track) as the cloud base height.

We have also added experiments about the effect of distance on R, RMSE and STD to prove the validity of this assumption. Please see the above parts of '*Regarding 1) Representativeness*' and '*Regarding 2) Algorithm development*' for detailed responses.

14. Section 3.3, line 172: Sentence should maybe be rephrased. I perceive that the authors take the $10^{th}$ percentile of the $H_{min}$ distribution for each 1° scene.

**RE:** This sentence has been revised as "Then, based on this $H_{min}$ data, we obtained the $10^{th}$ percentile of the $H_{min}$ distribution for each 1° scene as shown in Figure 3b" at Paragraph 1 in Section 3.2.

15. Figure 4, caption: A brief explanation for R, y, Matching data number, RMSE, STD should be provided.

**RE:** The caption of Figure 4 have been revised as follow:

**Figure 2:** (a) Scatter plot of CALIPSO CBH (the minimum CBH for each 1° scene) and ceilometer CBH at two marine sites in 2017. The triangle represents the data for the ENA site, and the crosses represent the data for the Barbados site. The color represents the shortest distance from the CALIPSO ground track to the ceilometer site. R is the Pearson correlation coefficient, y indicates the linear fitting relationship between ceilometer CBH and CALIPSO CBH, matching data number is the data amount of the scatter plot, RMSE is the root-mean-square error and STD is the standard deviation. (b) and (c) are the same as (a), but for CALIPSO CBH at 10 % quantile of $H_{min}$ and 20 % quantile of $H_{min}$, respectively. (d) Cumulative distribution of the difference between CALIPSO CBH and Ceilometer CBH at two sites in 2017.

16. Section 3.4.3, line 264: Rejection criterion (d) needs further elaboration. How do such cases compare with the ceilometer reference?

**RE:** Thank you for your comment. This criterion has now been removed, given its minimal impact on the retrieval results. In addition, we replaced it with the penetration efficiency of full-resolution clouds, the added content is as follows:

**Section 3.3.2:**

In addition, the penetration efficiency of full-resolution clouds is also taken into consideration, which is calculated using Eq. (3):

$$E_{lidar\_full} = N_{surface\_fullCloud}/N_{total\_fullCloud,} \tag{3}$$

$N_{surface\_fullCloud}$ refers to the number of CALIPSO lidar profiles that have both, a full-resolution clouds and a detectable surface; $N_{total\_fullCloud}$ is the total number of CALIPSO lidar profiles that detected the full-resolution clouds. The sensitivity test result of $E_{lidar\_full}$ is displayed in Figure 7.

We can see that the higher $E_{lidar\_full}$ the better cloud base height retrieval we can get, but when $E_{lidar\_full}$ is greater than 50 %, the amount of matched data is significantly reduced and the ability to retrieval high CBH is lost. Therefore, an optimal $E_{lidar\_full}$ of 50 % was chosen.

[Figure]

**Figure 7: Same as Fig. 5, but:** (a) when penetration efficiency of full-resolution cloud features ($E_{lidar\_full}$) of one scene is larger than 40 %; selection criteria of multilayer cloud fraction, and penetration efficiency of 333-m resolution cloud is the same as Figure 6c at two marine sites in 2017. The triangle represents the data for the ENA site, and the crosses represent the data for the Barbados site. The color represents the shortest distance from the CALIPSO ground track to the ceilometer site. (b) and (c) are the same as (a), but selection criteria of $E_{lidar\_full}$ in (b) is larger than 50 %, (c) is larger than 60 %, respectively. (d) Cumulative distribution of the difference between CALIPSO CBH and Ceilometer CBH at two sites in 2017. Different colored lines represent the cumulative distribution at different selection criteria of $E_{lidar\_full}$.

17. Section 4, line 276: Need to introduce "R" as correlation coefficient since it was not introduced before. Also state which kind is used, I am guessing Pearson correlation coeff.
    **RE:** The introduction of "R" is provided in the caption of Figure 4 as "R is the Pearson correlation coefficient".

18. Section 5, line 294: Or are the blanks also due to more cloud free scenes (e.g. Sahara, Australia)?
    **RE:** We have added this possible cause of the blanks as "because there are more scenes with cloud bases above 3 km or more cloud free scenes (e.g. Sahara, Australia)" at Paragraph 1 in Section 5.1.

19. Line 298: "The clouds with large CTHs are mainly distributed in the ocean area at low […] mainly over 2,000 m, which are consistent with the result of Sun-Mack et al." – change the beginning to: "Clouds with high CTHs occur mainly over ocean at low […]"; Further questions: "2,000 m" refer to surface height or CTHs? And what is consistent with Sun- Mack et al.?
    **RE:** The sentence has been revised as "Clouds with high CTHs occur mainly over ocean at low and high latitudes and over the land area, with CTHs over 2,000 m, which is consistent with the CTH result of Sun-Mack et al. (2014)" at Paragraph 2 in Section 5.1.

20. Line 304: "That will be helpful […]" – What exactly will be helpful is not clear.
    **RE:** The sentence at Paragraph 2 in Section 5.1 has been revised as "These shallow cloud geometric data retrieved in this study will be helpful …"

21. Figure 9 caption: Add that these are 2-year mean values.

**RE:** Figure 9 caption has been revised as "**Figure 3:** Geographic distributions of 2-year mean CBH, CTH and CGT on a 2° × 2° latitude–longitude grid in 2014 and 2017. The heights are in m above ground level."

22. Section 5.3., line 345: "More boundary layer clouds occur over land during the day-time than at nighttime" – maybe change to "Over land at middle and low latitudes, more boundary layer clouds are detected at day-time than at nighttime"

    **RE:** This sentence has been revised as you suggested at Paragraph 1 in Section 5.3.

23. Section 6, line 371: "because of the flow of warm continental air" – It is the subsidence of warm dry air at the subsiding branch of the Hadley Cell.

    **RE:** The tight relationships of CBH and CTH to the distance from land shows that our explanation is much more likely than a general subsidence, which is unaware of the position of the coastline.

24. Line 375: The CBH retrieval approach based on MISR observations (Böhm et al. 2019) is indeed limited by a 560m threshold height (over flat terrain). Below this height, no cloud retrievals are possible. The authors indicate here that this height limitation is at 700 m which should be clarified. The method by Böhm et al. has proven to work effectively for stratocumulus clouds of the southeast Pacific where the heights compare well to ground- based coastal observations (Munoz et al. 2016, https://doi.org/10.1175/JCLI-D-15-0757.1). However, for the southeast Atlantic the heights appear lower and the here proposed method is consistent with Andersen et al., 2019 (https://doi.org/10.5194/acp-19-4383-2019) (cf their Fig. 3c). Here, the MISR-based technique cannot capture the heights sufficiently.

    **RE:** The description of the comparison with MISR has been revised to "Therefore, these coastal regions are dominated by stratocumulus clouds, with CBH mainly below ~400 m and increasing up to ~1000 m far from the continent (the detailed distribution of CBH of the Southeast Atlantic is shown in Figure 15a), which is consistent with Andersen et al. (2019). The retrieved lower CBHs compared to previous study (Mülmenstädt et al., 2018) make it possible to estimate the coupling state and its relevance to the effects of aerosols on cloud fraction based on this dataset." in Section 6.

25. Line 380: MISR on the Terra platform has a 10a.m. equatorial overpass time whereas CALIPSO passes the equator in the afternoon. These diurnal differences should be kept in mind in particular in tropical regions, where heavy convection takes place in the afternoon. In general comparisons to other methods should always be seen in the light of obvious differences (e.g. overpass time, different years considered). Böhm et al. and Mülmenstädt et al. used 2007-2009 and 2007-2008, respectively, for the global assessment which is 5 to 10 years prior to 2014 and 2017 which are utilized here. Neither of these studies can claim to have an actual climatology as these periods are all far too short. It may be more appropriate to compare the resulting global averages to Mülmenstädt et al. as the same equatorial crossing times are given, and the methods are generally both based on CALIOP observations.

    **RE:** Thank you for your suggestion. We have modified the comparison in Section 6 with previous study to ensure similarity in overpass time and retrieval method, as follows. The retrieved lower CBHs compared to previous study (Mülmenstädt et al., 2018) make it possible to estimate the coupling state and its relevance to the effects of aerosols on cloud fraction based on this dataset.

26. Point v. Orographic clouds (line 391 ff): In particular for regions with complex orography, the assumption of homogeneous CBH across a larger region is most likely invalidated. Therefore, care should be taken when cloud heights are assessed for these regions. I suggest removing this conclusion here including Fig. 15 d.

    **RE:** Thank you for your suggestion. Content about Orographic clouds and Figure 15d have been

570         removed.

**Technical corrections**

1.  Line 32: "(3) CBH and CGT […] are 1200 and 1500 m, respectively" – ambiguous; does the 1200m refer to CBH for the Amazon and Congo, or for CBH and CGT for the Amazon?

    **RE:** This sentence has been revised as "CBH and CGT over the tropical rain forests (Amazon and Congo) are 1200 and 1500 m, respectively." in Abstract.

2.  Line 41: Sentence should be rephrased. Suggest to substitute "play a crucial role in the formation of size and concentration" by "and control size and number concentration"

    **RE:** This sentence has been revised as "Atmospheric aerosols, which serve as cloud condensation nuclei (CCN), control size and number concentration of cloud droplets and …" at Paragraph 1 in Introduction.

3.  Line 49: "It was shown recently […]" The meaning of this sentence is not clear to me, consider rephrasing.

    **RE:** This sentence has been revised as "Recent studies have shown that CGT can isolate the aerosol-cloud interaction from the influence of meteorology…" at Paragraph 2 in Introduction.

4.  Line 58: "wide-range" – no hyphen: "wide range"

    **RE:** "wide-range" has been revised as "wide range" at Paragraph 3 in Introduction.

5.  Line 61: "satellite observation data." – maybe just "satellite observations."

    **RE:** "satellite observation data" has been revised as "satellite observations" at Paragraph 3 in Introduction.

6.  Line 61: "The Suomi National […] (Baker, 2011)" – revise this sentence, maybe split into 2 sentences.

    **RE:** This sentence has been split into 2 sentences as "The Suomi National Polar-orbiting Partnership (Suomi NPP) Visible Infrared Imaging Radiometer Suite (VIIRS) retrieves CBH based on the CTH and CGT. However, VIIRS does not directly observe CGT, which is calculated by assuming $f_{ad} = 1$ (Baker, 2011)." at Paragraph 3 in Introduction.

7.  Line 65: "[…] and showed […]" – split sentence: "[…]. They showed […]"

    **RE:** This sentence has been split into 2 sentences as you suggested at Paragraph 3 in Introduction.

8.  Line 68: "Li et al. […] CloudSat is 540 m" – rephrase; do you mean, they compared their retrieval to CloudSat CBH retrieval and found a standard deviation of 540m?

    **RE:** This sentence has been revised as "By comparing their retrieval to CloudSat CBH retrieval, a standard deviation of 540 m was found." at Paragraph 3 in Introduction.

9.  Line 75: "have a good potential for retrieval" – substitute with "have the potential for accurate retrieval"

    **RE:** "have a good potential for retrieval" has been revised as "have the potential for accurate retrieval" at Paragraph 4 in Introduction.

10. Line 87: "[…] we proposed in this study […]" – substitute with "we derived"

    **RE:** "we proposed in this study" has been revised as "we derived" at Paragraph 5 in Introduction.

11. Line 88: "retrieve the global CBH […]" – modify "retrieve the global distribution of CBH […]"

    **RE:** "retrieve the global CBH" has been revised as "retrieve the global distribution of CBH" at Paragraph 5 in Introduction.

12. Line 89: "The low-level clouds […] aerosol-cloud interaction research." – additional motivation; consider moving to a previous paragraph.

**RE:** This sentence in Paragraph 5 been moved to Paragraph 4 in Introduction.

13. Line 93: "CALIPSO-retrieved CBHs were validated against in situ ceilometer measurements in Section 4." – I think, generally present tense should be used ("were" -> "are") and change "in Section 4." to "(Section 4)."

   **RE:** This sentence has been revised as "The CALIPSO-retrieved CBHs are evaluated and validated against in situ ceilometer measurements (Section 4)." at Paragraph 5 in Introduction.

14. Line 93: "Based on the validated-CBH, CTH and CGT were retrieved globally […]" – no hyphen; also not clear if the CTH and CGT is based on the CBH? How do you mean?

   **RE:** This sentence has been revised as "Based on the validated CBH, we retrieved CTH and CGT globally and produced global annual, seasonal and diurnal distributions maps of CBH, CTH and CGT (Section 5)." at Paragraph 5 in Introduction.

15. Line 95: "in Section 5." – shorten to "(Section 5)."

   **RE:** "in Section 5." has been revised as "(Section 5)." at Paragraph 5 in Introduction.

16. Line 95: "Several features" – maybe change to "Specific spatial patterns"

   **RE:** "Several features" has been revised as "Specific spatial patterns" at Paragraph 5 in Introduction.

17. Line 95: "based on this high-precision cloud geometry information." – remove this phrase

   **RE:** This phrase has been removed.

18. Line 135: "The objective of CBH retrieval is to retrieve the forming level of the clouds" –change to "The objective of the CBH retrieval is to retrieve the forming level of clouds"

   **RE:** This sentence has been revised as you suggested at Paragraph 1 in Section 3.

19. Line 141: "[…] of CALIPSO VFM scenes that identified low-level water clouds." – change to "for CALIPSO VFM scenes which are identified as low-level water clouds."

   **RE:** This sentence has been revised as you suggested at Paragraph 1 in Section 3.1.

20. Line 178: Sentence should be revised. Maybe start with "Extremely low $H_{min}$ are more prone to misclassification because […]"

   **RE:** This sentence has been revised as "Extremely low $H_{min}$ are more prone to misclassification because mixture of surface signals and cloud features or VFM misclassified near surface thick aerosols as clouds …"at Paragraph 2 in Section 3.2.

21. Line 182: "which the center point to the ceilometer observation station within a distance of 150 km" – is missing a verb.

   **RE:** This sentence has been revised as "Then we obtained the distribution of the base height of cloud features observed by ceilometers within 30 minutes before and after the CALIPSO overpass time ($CB_{ceilo}$)" in Section 2.3.

22. Line 211: "collected in that given […]" – change to "collected in a given […]"

   **RE:** "collected in that given" has been revised as "collected in a given" at Paragraph 1 in Section 3.3.1.

23. Line 243: "$E_{lidar}$ was used to […]" – consider revising the sentence, maybe: "$E_{lidar}$ is used to determine the lowest penetration efficiency that can still provide valid cloud base information in this study, […]"

   **RE:** This sentence has been revised as "$E_{lidar}$ is used to determine the lowest penetration efficiency that can still provide valid cloud base information of 333-m resolution cloud in this study, …" at Paragraph 1 in Section 3.3.2.

24. Line 246: "[…] that both have […]" – change to "that have both, a 333 m […] and a detectable

surface […]"

**RE:** This sentence has been revised as "$N_{surface\_333}$ refers to the number of CALIPSO lidar profiles that have both, a 333-m horizontal resolution clouds and a detectable surface…" at Paragraph 1 in Section 3.3.2.

25. Line 292: "in 2014 were also applied in this study to ensure most the grids" – change to "for 2014 were also applied in this study to ensure that most of the grids"

**RE:** This sentence has been removed at Paragraph 1 in Section 5.1.

26. Line 295: "The CBH distribution (Figure 9a) shows most CBHs above surface in the land area are higher than over ocean." – change to "The distribution of CBH above ground level (Figure 9a) shows that over land, CBHs are higher than over ocean."

**RE:** This sentence has been revised as you suggested at Paragraph 2 in Section 5.1.

27. Line 300: "smallest" – change to "lowest"; "areas" – change to "regions"

**RE:** This sentence has been revised as "the lowest CTHs which are mainly ~1,000 m are also concentrated in offshore regions …" at Paragraph 2 in Section 5.1.

28. Line 301: "which are mainly ~1,000m" – move next to CTH if it refers to that.

**RE:** This sentence has been revised as "the lowest CTHs which are mainly ~1,000 m are also concentrated in offshore regions and the equatorial regions of the western hemisphere, in agreement with (Zuidema et al. (2009))." at Paragraph 2 in Section 5.1.

29. Line 302: "Thus, shallow clouds with small CGTs (<800 m) mainly distributed at the mid latitude oceanic area and offshore areas with a percentage of ~10 %." – change to "Thus, shallow clouds with small CGTs (<800 m) occur mainly over mid latitude oceanic regions and eastern margins of subtropical oceans with a percentage of ~10 %."; What do the 10% refer to, cloud cover overall or portion of shallow low clouds?

**RE:** This sentence has been revised as "Thus, shallow clouds with small CGTs (<800 m), with a percentage of ~10% of all low-level clouds, occur mainly over mid latitude oceanic regions and eastern margins of subtropical oceans." at Paragraph 2 in Section 5.1.

30. Line 336: "land than over the ocean […]" – change to "land than over ocean"

**RE:** "land than over the ocean" has been revised as "land than over ocean" at Paragraph 2 in Section 5.2.

31. Line 407: missing a verb

**RE:** This sentence has been revised as "At the same time, we used 1° scene along the CALIPSO track for CBH retrieval. In addition, the 10 % percentile of all cloud base information, and the 40 % quantile of the first local peak was used as initial CBHs for over the ocean and land, respectively. All of these operations can reduce the effect of cloud anvils on the CALIPSO retrieval of CBH to some extent. The methodology was developed based on observations for the year 2017 from two ocean ceilometer stations." at Paragraph 1 in Conclusions.

32. Line 408: "was tested and validated based on two in situ ceilometer measurements in 2017" – change to "was tested and validated based on observations for the year 2017 from two ceilometer stations"

**RE:** This sentence has been revised as "The methodology was developed based on observations for the year 2017 from two ocean ceilometer stations." at Paragraph 1 in Conclusions.

---

## Author Comment (AC2)

Atmos. Chem. Phys. Discuss., referee comment RC2 https://doi.org/10.5194/acp-2020-1252-RC2, 2021

5 © Author(s) 2021. This work is distributed under the Creative Commons Attribution 4.0 License.

**10 **Comment on acp-2020-1252**

**Anonymous Referee #2**

Referee comment on "Satellite retrieval of cloud base height and geometric thickness of low-level cloud based on CALIPSO" by Xin Lu et al., Atmos. Chem. Phys. Discuss., https://doi.org/10.5194/acp-2020-1252-RC2, 2021

Lu et al. "Satellite retrieval of cloud base height and geometric thickness of low-level cloud based

20 on CALIPSO"

15

25

This study proposes a method to retrieve the cloud base height from CALIPSO observations, from which cloud thickness is further derived. It also shows the statistical results of cloud macrophysical properties based on their retrievals. In principle, the study are interesting, while I have some concerns as listed below. Thus, I would recommend its acceptance for publication after necessary modifications.

**General comments**

**Question 1:**

While this study shows the cloud base properties from CALIPSO, the representation of the statistical properties should be discussed. Particularly, only partial clouds over the world have been retrieved with the proposed method, how could people use the statistical information obtained to represent all cloud properties? In addition, the cloud bases along with the statistical values from CALIPSO should be evaluated with the CloudSat. Particularly, using only two ground station observation (also not long-term observations) to evaluate the performance of the method seems to me not sufficient.

**RE:** Thank you for your comment.**

We qualify that the cloud base heights only for the conditions satisfying the selection criteria. This means mostly broken or thin boundary layer clouds. We have added the description in the paper.

**Paragraph 4 in Section 3.3.2:**

40 After the above processing, we obtained the final CBH of that CALIPSO 1° scene. These cloud base height information are mainly extracted from broken or thin boundary layer clouds.

Moreover, CloudSat has difficulties for retrieving CBH of low-level clouds for the following reasons:

1. The ground clutter prevents CloudSat detection of very low cloud base.

- 2. Rain from precipitating clouds produces radar returns below cloud base.
- 3. Due to the dependence of radar reflectivity on the 6th power of cloud droplet diameter, the reflectivity of clouds with small droplets can be below the CloudSat minimum detectable signal, especially near cloud base where cloud droplets are smallest.

We have also added this description to Paragraph 3 in Introduction in the manuscript.

Furthermore, to validate the applicability over land of the retrieval algorithm in this study and to increase the case number for validation, we conducted additional validation experiments using 4 50 years of ceilometer data from 138 continental sites in the Great Plains of the USA (Figure R1, this figure is also provided as Figure A2 in the Appendix of the paper). These ceilometer data are derived from Automated Surface Observation System (ASOS). The data source is https://mesonet.agron.iastate.edu/request/download.phtml (last access: May 2021), which is maintained by Iowa Environmental Mesonet of Iowa State University. The data period used in this 55 study is taken from 2017 to 2020 because during this period, the ceilometers provide better time resolution of cloud base measurements.

Figure R1. The ASOS ceilometer distribution over land used for CBH evaluation.

60 The ASOS uses a laser beam ceilometer with a time interval of 5-30 minutes, with a vertical resolution of ~30 m, and a vertical detection range of ~3700 m. In order to ensure the data quality of the ASOS ceilometer observations, we only use ceilometer data with CBH less than 3000 m and limit the number of valid cloud observations to no less than 2 during 1 hour.

Then, based on the CBH retrieval algorithm developed from the cloud observations over ocean, we conducted the same experiment on land to test the applicability of our algorithm. Since the cloud base is not as homogeneous over land as over ocean, we consider using the cloud information below the first peak nearest to the surface in the cloud fraction profile of 1° scene as a proxy for all cloud base information in this scene (which we defined as the first local peak above surface). In this way, we avoid missing the newly developed clouds with small size. Therefore instead of using Hmin at 10 %

70 quantile of all clouds as the initial CBH over ocean, we tested the CBH at different quartiles of the first local peak as the initial CBH for the scene. Table R1 shows the corresponding statistical results (this table is also provided as Table B1 in the Appendix of the paper), which can be seen there is a minimum RMSE when the initial CALIPSO CBH is at the 40% quantile of the first local peak which closest to surface. This has also been verified in the results of other years (2007-2016, not shown).

45

75 Thus on land the CALIPSO-retrieved initial CBH is 40% quantile of first local peak, while over the ocean it is  $H_{min}$  at 10 % quantile.

**Table R1.** Validation statistics of CALIPSO-retrieved initial CBH for different conditions of 138 continental ceilometer sites in the southern Great Plains in 2017-2020.

|                                                   | Case number | RMSE (m) |
|---------------------------------------------------|-------------|----------|
| min CALIPSO CBH                                   | 8404        | 635      |
| H min at 10 % quantile (same as ocean) | 7963        | 665      |
| 10 % quantile of the first local peak             | 8302        | 616      |
| 20 % quantile of the first local peak             | 8280        | 615      |
| 30 % quantile of the first local peak             | 8270        | 619      |
| 40 % quantile of the first local peak             | 8203        | 612      |
| 50 % quantile of the first local peak             | 8225        | 618      |
| 60 % quantile of the first local peak             | 8193        | 619      |
| 70 % quantile of the first local peak             | 8169        | 628      |

Then, based on the CBH data obtained from the above processing, we further tested the effects of Fmulti, Elidar and Flidar\_full over land following the same process as over ocean, as showed in Figure R2 (this figure is also provided as Figure A3 in the Appendix of the manuscript). From the results it can be seen that the optimal thresholds for Elidar>50%, and Flidar\_full>50% over land are consistent with those over ocean, which also shows that the CBH retrieval algorithm we developed based on cloud observations from the ocean is applicable on land. The parameters Fmulti<40% has little meaning over land due to the scarcity of the cases that fulfil that condition. It is kept over land for consistency with the ocean criteria.</li>

Figure R2. (a) Joint distribution of CBH absolute error (between CALIPSO CBH and ceilometer CBH) and multilayer cloud fraction of 138 continental ceilometer sites in the southern Great Plains in 2017-2020 with distance less than 50 km. The different coloured line represents at different time (the black line: all time; the red line: day-time; the blue line: night-time). The I-beam line represents the standard error. (b) is the same as (a), but for penetration efficiency of 333-m resolution cloud.

(c) is the same as (a), but for penetration efficiency of all cloud.

95

As mentioned before due to the complexity of topography and land surface situation, the cloud bases height varies at larger spatial scales. The 150 km distance between the shortest distance from the CALIPSO ground track to the ceilometer site cannot be used for over land validation. We have to shrink the distance to minimize the spatial variability due to the changes over land. We tested the effect of distance (that is the shortest distance from the CALIPSO ground track to the ceilometer site) and observation time on the retrieval results (Figure R3, which is also provided as Figure A4 in the

- Appendix of the manuscript). The results (Figure R3b) show that the absolute error between the CALIPSO CBH and the ceilometer CBH becomes smaller as the distance decreases, stabilizing at distances less than 50 km. It is therefore preferable to limit the distance to 50 km for studies on land to better meet the assumptions of a homogeneous CBH within 1° scenes. It can also be seen that the cloud base heights are more evenly distributed during the day-time (Figure R3a, 300-1800 m) than at night, while at night CBHs are mainly concentrated below about 700 m and are most frequent very near the surface, where validation becomes unreliable. In addition to the distance limitation, the cloud base homogeneity further constrained by comparing the lifted condensation level (HLCL) to
- ceilometer cloud base height. To satisfy the cloud base homogeneity assumption (Efraim et al., 2020), cases are selected when the absolute difference between the  $H_{LCL}$ (calculated from ASOSobserved air temperature and dew point temperature) and ceilometer CBH is less than 200 m. In summary, the ceilometer measurements need to satisfy the following conditions for validating CALIPSO CBH retrieval: 1) The ceilometer is within 50 km radius to the center of CALIPSO ground track; 2) the ceilometer measured CBH should have an absolute difference less than 200 m against  $H_{LCL}$  as calculated from surface measured air temperature and dew point.
- 115

---

## Author Response (AR2)

**Title:** Satellite retrieval of cloud base height and geometric thickness of low-level cloud based on CALIPSO

**Author(s):** Xin Lu, Feiyue Mao, Daniel Rosenfeld, Yannian Zhu*, Zengxin Pan, and Wei Gong

**MS No.:** acp-2020-1252

**Iteration:** Minor Revision

Dear Prof. & Dr. Johannes Quaas,

Very appreciate for your and the reviewer's comments. We have revised our manuscript in light of all the comments. We hope our revisions have addressed the raised issues. The changes have been highlighted in color in the manuscript, and summarized as follows.

1. In Introduction, we have revised the relevant descriptions and added references in accordance with the reviewer's comments.

2. We have explained and revised the descriptions that caused the misunderstanding in Sections 2.3 and 3.1.

3. The fitting regression equations in Figures 4-8 have been revised in the manuscript.

Please refer to changes in line with the reviewer's comments for more details. Thank you for your time.

Sincerely,

Xin and Yannian

**Editor Decision**:

Publish subject to minor revisions (review by editor) (14 Jun 2021) by Johannes Quaas

25 **Comments to the Author:**

Dear authors,

Reviewer #2 has a rather limited number of minor suggestions that I ask you to address in a (probably last) iteration.

Best regards,

30 Johannes

**Changes in Line with the Comments of Referee 2**

**At first, very appreciate for your insightful comments and suggestions. We have revised our paper in light of your comments. The changes have been highlighted in color in the manuscript, and are summarized below.**

**General comments**

Lu et al. "Satellite retrieval of cloud base height and geometric thickness of low-level cloud based on CALIPSO"

The authors have solved most of my concerns. I would recommend its acceptance for publication after some minor revisions.

**Minor comments:**

1. Line 42-43, For "The combination of cloud base updraft and CCN determines cloud base droplet concentration ($N_d$)", I wonder if this is correct since I think the cloud base droplet concentration should also be dependent on the supersaturation, wind shear, and so on.

   **RE:** Thank you for your comment. The description here has been revised as following:

   **Paragraph 1 in Introduction:**

   The cloud base droplet concentration ($N_d$) is determined by the cloud base updraft, CCN, supersaturation, wind shear, and so on, which in turn determines the cloud's albedo for a given liquid water path (Twomey, 1974;Sato and Suzuki, 2019).

2. Line 50-52, A recent study has also indicated this point and tried to improve the simulations of cloud cover, base and top heights, which is worthy to cite here too, Ma et al. (2018, Doi: 10.1002/2017MS001234).

   **RE:** Thank you for your suggestion. This reference has been added to the citation at Paragraph 2 in Introduction as "CBH and CTH are fundamental cloud properties that are required to be parameterized correctly for improving model simulations of climate and climate change (Grosvenor et al., 2017;Zhao and Suzuki, 2019;Lenaerts et al., 2020;Ma et al., 2018)".

3. Line 59-60, This is true for active remote sensing of satellites. For passive remote sensing of satellites, the CTHs could also be relatively easily obtained, while likely with large uncertainties.

   **RE:** Thank you for your comment. The description here has been revised as following:

   **Paragraph 3 in Introduction:**

   Although there is often a large uncertainty in the cloud top heights obtained from passive satellite observations, it is relatively simple to retrieve. In contrast, the retrieval of CBH is much more challenging but necessary for retrieving CGT.

4. Line 143-144, While I do not understand the rationale for "The lowest 10 % quantile of the $CB_{ceilo}$ is determined as true CBH", I think it might be reasonable considering that Ceilometer often overestimate the cloud base height as indicated by Wang et al. (2018, Doi: 10.1016/j.atmosres.2017.11.021). If the authors could explain the rationale to me, that would be great. I also suggest the authors could use the study mentioned here to support the method used.

   **RE:** We use the lowest 10 % quantile of the $CB_{ceilo}$ rather than the mean or minimum of the $CB_{ceilo}$ as true CBH, because the mean is highly susceptible to the influence of developing higherlevel clouds and the minimum may be influenced by instrument noise. In addition, the 10% quantile used is also consistent with the quantile used in this study for the retrieval of cloud base heights based on CALIPSO satellite observations.

Accordingly, the description here also has been revised as you suggested:

**Paragraph 1 in Section 2.3:**

To avoid the underestimation of low CBHs and overestimation of high CBHs by ceilometer due to the influence of developing higher-level clouds and ceilometer measurement noise, the lowest 10 % quantile of the $CB_{ceilo}$ is determined as true CBH (Wang et al., 2018).

5. Line 179-181, you may revise the sentence to state more clearly. Based on the descriptions before this sentence, I would say a resolution higher than 333 m could be better. I think what you would like to say is "compared to previous studies based on coarse spatial resolution satellite data".

**RE:** Sorry for that confusion. 333 m is the highest resolution of the CALIPSO satellite as shown in Figure 2b. We have added the relevant description as following:

**Paragraph 1 in Section 2.1:**

The VFM also provides the thermodynamic phase of cloud layers (water cloud, ice cloud) and horizontal resolution (333 m, 1 km, 5 km, 20 km, 80 km) that the retrieval was based on.

Accordingly, the description here has been revised as you suggested to make it more clearly:

**Paragraph 2 in Section 3.1:**

Therefore, it is more reliable to use the water cloud information with the highest resolution of 333 m to retrieve the CBH of low-level clouds compared to previous studies based on coarse spatial resolution CALIPSO satellite data.

6. Figure 4 d) The xtitle is necessary.

**RE:** Thank you for your suggestion. We have added the xtitle of Figure 4d as shown below.

[Figure]

**Figure 4: (a) Scatter plot of CALIPSO CBH (the minimum CBH for each 1° scene) and ceilometer CBH at two marine sites in 2017.** The triangle represents the data for the ENA site, and the crosses represent the data for the Barbados site. The color represents the shortest distance from the CALIPSO ground track to the ceilometer site. R is the Pearson correlation coefficient, y indicates the linear fitting relationship between ceilometer CBH and CALIPSO CBH, matching data number is the data amount of the scatter plot, RMSE is the root-mean-square error and STD is the standard deviation. (b) and (c) are the same as (a), but for CALIPSO CBH at 10 % quantile of $H_{min}$ and 20 % quantile of $H_{min}$, respectively. (d) Cumulative distribution of the difference between CALIPSO CBH and Ceilometer CBH at two sites in 2017.

7. Line 223, ":" -> "."

   **RE:** This ":" has been revised as "." at Paragraph 1 in Section 3.3.

8. Figure 8, What are the fitting regression equations shown in the figures? They are too weird to me with so large slope and intercept values.

   **RE:** Thank you for pointing it out. Yes, the fitting regression equation here was incorrect, we have corrected it in Figure 8 as showed below. Also other fitting regression equations have been updated (Figures 4, 5, 6, and 7) in the manuscript.

[Figure]

**Figure 8. Validation of CALIPSO-retrieved CBH against 138 continental ceilometer sites in the southern Great Plains in 2017-2020.** (a) Scatter plot of CALIPSO CBH and ceilometer CBH on all-time. The color represents the shortest distance from the CALIPSO ground track to the ceilometer site. (b) Cumulative distribution of the difference between CALIPSO CBH and Ceilometer CBH on all-time. (c)/(e) and (d)/(f) are the same as (a) and (b), but for day-time and night-time, respectively.

**Reference**

Grosvenor, Daniel, P., Field, Paul, R., Hill, Adrian, A., and Shipway: The relative importance of macrophysical and cloud albedo changes for aerosol-induced radiative effects in closed-cell stratocumulus: insight from the modelling of a case study, Atmos. Chem. Phys. 2017.

Lenaerts, J. T., Gettelman, A., Van Tricht, K., van Kampenhout, L., and Miller, N. B.: Impact of Cloud Physics on the Greenland Ice Sheet Near-Surface Climate: A Study With the Community Atmosphere Model, Journal of Geophysical Research: Atmospheres, 125. 2020.

Ma, Z., Liu, Q., Zhao, C., Shen, X., Wang, Y., Jiang, J. H., Li, Z., and Yung, Y.: Application and Evaluation of an Explicit Prognostic Cloud-Cover Scheme in GRAPES Global Forecast System, Journal of Advances in Modeling Earth Systems, 10, 652-667. doi:https://doi.org/10.1002/2017MS001234, 2018.

Sato, Y., and Suzuki, K.: How do aerosols affect cloudiness?, Science, 363, 580-581. 2019.

Twomey, S.: Pollution and the planetary albedo, Atmospheric Environment (1967), 8, 1251-1256. 1974.

Wang, Y., Zhao, C., Dong, Z., Li, Z., Hu, S., Chen, T., Tao, F., and Wang, Y.: Improved retrieval of cloud base heights from ceilometer using a non-standard instrument method, Atmospheric Research, 202, 148-155. 2018.

Zhao, S., and Suzuki, K.: Differing Impacts of Black Carbon and Sulfate Aerosols on Global Precipitation and the ITCZ Location via Atmosphere and Ocean Energy Perturbations, Journal of Climate, 32. 2019.